# Joint Bayesian Inference of Graphical Structure and Parameters with a Single Generative Flow Network

**Tristan Deleu**[1]   **Mizu Nishikawa-Toomey**[1]   **Jithendaraa Subramanian**[2]
**Nikolay Malkin**[1]   **Laurent Charlin**[3]   **Yoshua Bengio**[1,4]

Mila – Quebec AI Institute

## Abstract

Generative Flow Networks (GFlowNets), a class of generative models over discrete and structured sample spaces, have been previously applied to the problem of inferring the marginal posterior distribution over the directed acyclic graph (DAG) of a Bayesian Network, given a dataset of observations. Based on recent advances extending this framework to non-discrete sample spaces, we propose in this paper to approximate the joint posterior over not only the structure of a Bayesian Network, but also the parameters of its conditional probability distributions. We use a single GFlowNet whose sampling policy follows a two-phase process: the DAG is first generated sequentially one edge at a time, and then the corresponding parameters are picked once the full structure is known. Since the parameters are included in the posterior distribution, this leaves more flexibility for the local probability models of the Bayesian Network, making our approach applicable even to non-linear models parametrized by neural networks. We show that our method, called JSP-GFN, offers an accurate approximation of the joint posterior, while comparing favorably against existing methods on both simulated and real data.

## 1   Introduction

As a compact representation for complex probabilistic models, Bayesian Networks are a framework of choice in many fields, such as computational biology (Friedman et al., 2000; Sachs et al., 2005) and medical diagnosis (Lauritzen and Spiegelhalter, 1988). When the directed acyclic graph (DAG) structure of the Bayesian Network—which specifies the possible conditional dependences among the observed variables—is known, it can be used to perform probabilistic inference for queries of interest with a variety of exact or approximate methods (Koller and Friedman, 2009). However, if this graphical structure is unknown, one may want to infer it based on a dataset of observations $\mathcal{D}$.

In addition to being a challenging problem due to the super-exponentially large search space, learning a single DAG structure from data may also lead to confident but incorrect predictions (Madigan et al., 1994), especially in cases where the evidence is limited. In order to avoid misspecifying the model, it is therefore essential to quantify the epistemic uncertainty about the structure of the Bayesian Network. This can be addressed by taking a Bayesian perspective on structure learning and inferring the posterior distribution $P(G \mid \mathcal{D})$ over graphs given our observations. This (marginal) posterior can be approximated using methods based on Markov chain Monte Carlo (MCMC; Madigan et al., 1995) or variational inference (Cundy et al., 2021; Lorch et al., 2021). However, all of these methods rely on the computation of the marginal likelihood $P(\mathcal{D} \mid G)$, which can only be done efficiently in closed form for limited classes of models, such as linear Gaussian (Geiger and Heckerman, 1994),

---

[1]Université de Montréal, [2]McGill University, [3]HEC Montréal, [4]CIFAR Senior Fellow.
Correspondence: Tristan Deleu (deleutri@mila.quebec)

Code available at: https://github.com/tristandeleu/jax-jsp-gfn

37th Conference on Neural Information Processing Systems (NeurIPS 2023).

discrete models with Dirichlet prior (Heckerman et al., 1995), or non-linear models parametrized with a Gaussian Process (von Kügelgen et al., 2019).

While there exists a vast literature on Bayesian structure learning to approximate the marginal posterior distribution, inferring the *joint posterior* $P(G, \theta \mid \mathcal{D})$ over both the DAG structure $G$ of the Bayesian Network and the parameters $\theta$ of its conditional probability distributions—the probability of each variable given its parents—has received comparatively little attention. The main difficulty arises from the mixed sample space of the joint posterior distribution, with both discrete components (the graph $G$) and continuous components (the parameters $\theta$), where the dimensionality of the latter may even depend on $G$. However, modeling the posterior distribution over $\theta$ has the notable advantage that the conditional probability distributions can be more flexible (e.g., parametrized by neural networks): in general, computing $P(\mathcal{D} \mid G, \theta)$ is easier than computing the marginal $P(\mathcal{D} \mid G)$, lifting the need to perform intractable marginalizations.

Since they provide a framework for generative modeling of discrete and composite objects, Generative Flow Networks (GFlowNets; Bengio et al., 2021, 2023) proved to be an effective method for Bayesian structure learning. In Deleu et al. (2022), the problem of generating a sample DAG from the marginal posterior $P(G \mid \mathcal{D})$ was treated as a sequential decision process, where edges are added one at a time, starting from the empty graph over $d$ variables, following a learned transition probability. Nishikawa-Toomey et al. (2023) have also proposed to use a GFlowNet to infer the joint posterior $P(G, \theta \mid \mathcal{D})$; however, they used it in conjunction with Variational Bayes to update the distribution over $\theta$, getting around the difficulty of modeling a continuous distribution with a GFlowNet.

In this paper, we propose to infer the joint posterior over graphical structures $G$ and parameters of the conditional probability distributions $\theta$ of a Bayesian Network using *a single* GFlowNet called JSP-GFN (for *Joint Structure and Parameters GFlowNet*), leveraging recent advances extending GFlowNets to continuous sample spaces (Lahlou et al., 2023), and expands the scope of applications for Bayesian structure learning with GFlowNets. The generation of a sample $(G, \theta)$ from the approximate posterior now follows a two-phase process, where the DAG $G$ is first constructed by inserting one edge at a time, and then the corresponding parameters $\theta$ are chosen once the structure is completely known. To enable efficient learning of the sampling distribution, we introduce new conditions closely related to the ones derived in Deleu et al. (2022), based on the subtrajectory balance conditions (Malkin et al., 2022), and show that they guarantee that the GFlowNet does represent $P(G, \theta \mid \mathcal{D})$ once they are completely satisfied. We validate empirically that JSP-GFN provides an accurate approximation of the posterior when those conditions are approximately satisfied by a learned sampling model, and compares favorably against existing methods on simulated and real data.

## 2  Background

**Notations.**  Throughout this paper, we will work with directed graphs $G = (V, E)$, where $V$ is a set of nodes, and $E \subseteq V \times V$ is a set of (directed) edges. For a node $X \in V$, we denote by $\mathrm{Pa}_G(X)$ the set of parents of $X$ in $G$, and $\mathrm{Ch}_G(X)$ the set of its children. For two nodes $X, Y \in V$, $X \to Y$ represents a directed edge $(X, Y) \in E$ (denoted $X \to Y \in G$), and $X \rightsquigarrow Y$ represents a directed path from $X$ to $Y$, following the edges in $E$ (denoted $X \rightsquigarrow Y \in G$).

In the context of GFlowNets (see Section 2.2), an *undirected path* in a directed graph $\mathcal{G} = (\mathcal{V}, \mathcal{E})$ between two states $s_0, s_n \in \mathcal{V}$ is a sequence of vertices $(s_0, s_1, \ldots, s_n)$ where either $(s_i, s_{i+1}) \in \mathcal{E}$ or $(s_{i+1}, s_i) \in \mathcal{E}$ (i.e., following the edges of the graph, regardless of their orientations).

### 2.1  Bayesian structure learning

A Bayesian Network is a probabilistic model over $d$ random variables $\{X_1, \ldots, X_d\}$, whose joint distribution factorizes according to a directed acyclic graph (DAG) $G$ as

$$P(X_1, \ldots, X_d \mid \theta, G) = \prod_{i=1}^{d} P\big(X_i \mid \mathrm{Pa}_G(X_i); \theta_i\big), \tag{1}$$

where $\theta = \{\theta_1, \ldots, \theta_d\}$ represents the parameters of the conditional probability distributions (CPDs) involved in this factorization. When the structure $G$ is known, a Bayesian Network offers a representation of the joint distribution that may be convenient for probabilistic inference (Koller and

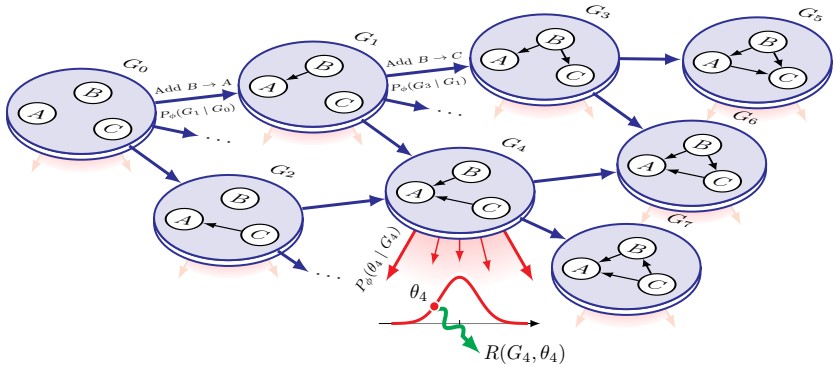

Figure 1: Structure of the Generative Flow Network to approximate the joint posterior distribution $P(G, \theta \mid \mathcal{D})$. A graph $G$ and parameters $\theta$ are constructed as follows: starting from the empty graph $G_0$, (1) the graph $G$ is first generated one edge at a time (blue), as in (Deleu et al., 2022). Then once we select the action indicating that we stop adding edges to the graph, (2) we generate the parameters $\theta$ (red), conditioned on the graph $G$. Finally, given $G$ and $\theta$, (3) we receive a reward $R(G, \theta)$ (green).

Friedman, 2009), as well as learning its parameters $\theta$ using a dataset $\mathcal{D} = \{\boldsymbol{x}^{(1)}, \ldots, \boldsymbol{x}^{(N)}\}$, where each $\boldsymbol{x}^{(j)}$ represents an observation of the $d$ random variables $\{X_1, \ldots, X_d\}$.

However, when the structure of the Bayesian Network is unknown, we can also *learn* the DAG $G$ from data (Spirtes et al., 2000; Chickering, 2002). Using a Bayesian perspective, we may want to either model the (marginal) posterior distribution $P(G \mid \mathcal{D})$ over only the DAG structures, or the joint posterior $P(G, \theta \mid \mathcal{D})$ over both the structure $G$, as well as the parameters of the CPDs $\theta$.

## 2.2 Generative Flow Networks

A *Generative Flow Network* (GFlowNet; Bengio et al., 2021, 2023) is a generative model over a structured sample space $\mathcal{X}$. The structure of the GFlowNet is described by a DAG $\mathcal{G}$ whose vertex set is the *state space* $\mathcal{S}$, where $\mathcal{X} \subseteq \mathcal{S}$. This should not be confused with the DAG in a Bayesian Network: in fact, each state in $\mathcal{G}$ could itself represent the structure of a Bayesian Network (Deleu et al., 2022). A sample $x \in \mathcal{X}$ is constructed sequentially by following the edges of $\mathcal{G}$, starting at a special initial state $s_0$, until we reach the state $x$. We define a special terminal state $s_f$, a transition to which indicates the end of the sequential process. The states in $\mathcal{X}$ are those for which there is a directed edge $x \to s_f$; they are called *complete states*[1] and correspond to valid samples of the distribution induced by the GFlowNet. A path $s_0 \rightsquigarrow s_f$ in $\mathcal{G}$ is called a complete trajectory. An example of the structure of a GFlowNet is given in Section 3.1 and Figure 1.

Every complete state $x \in \mathcal{X}$ is associated with a *reward* $R(x) \geq 0$, indicating the unnormalized probability of $x$. We use the convention $R(s) = 0$ for any state $s \in \mathcal{S} \backslash \mathcal{X}$, since they do not correspond to valid samples of the distribution. Bengio et al. (2021) showed that if there exists a function $F_\phi(s \to s') \geq 0$ defined over the edges of $\mathcal{G}$, called a *flow*, that satisfies the following *flow-matching conditions*

$$\sum_{s \in \text{Pa}_{\mathcal{G}}(s')} F_\phi(s \to s') - \sum_{s'' \in \text{Ch}_{\mathcal{G}}(s')} F_\phi(s' \to s'') = R(s') \tag{2}$$

for all the non-terminal states $s' \in \mathcal{S}$, then the GFlowNet induces a distribution over complete states proportional to the reward. More precisely, starting from the initial state $s_0$, if we sample a complete trajectory $(s_0, s_1, \ldots, s_{T-1}, x, s_f)$ following the forward transition probability, defined as

$$P(s_{t+1} \mid s_t) \propto F_\phi(s_t \to s_{t+1}), \tag{3}$$

with the conventions $s_T = x$ and $s_{T+1} = s_f$, then the marginal probability that a trajectory sampled following $P$ terminates in $x \in \mathcal{X}$ is proportional to $R(x)$. The flow function $F_\phi(s \to s')$ may be parametrized by a neural network and optimized to minimize the error in (2), yielding a transition model that can be used to approximately sample from the distribution on $\mathcal{X}$ proportional to $R$.

---

[1]Bengio et al. (2023) also call these states *terminating*; we follow the naming conventions of Deleu et al. (2022) here to avoid ambiguity, as it is closely related to our work.

## 2.3 Structure learning with GFlowNets

Since GFlowNets are particularly well-suited to specifying distributions over composite objects, Deleu et al. (2022) used this framework in the context of Bayesian structure learning to approximate the (marginal) posterior distribution over DAGs $P(G \mid \mathcal{D})$. Their model, called *DAG-GFlowNet*, operates on the state-space of DAGs, where each graph is constructed sequentially by adding one edge at a time, starting from the empty graph with $d$ nodes, while enforcing the acyclicity constraint at every step of the generation (i.e., an edge is not added if it would introduce a cycle). Its structure is illustrated at the top of Figure 1, where it forms the part of the graph shown in blue.

Instead of working with flows $F_\phi(G \to G')$, as in Section 2.2, DAG-GFlowNet directly learns the forward transition probability $P_\phi(G' \mid G)$ that satisfies the following alternative *detailed balance* conditions for any transition $G \to G'$ (i.e., $G'$ is the result of adding a single edge to $G$):

$$R(G')P_B(G \mid G')P_\phi(s_f \mid G) = R(G)P_\phi(G' \mid G)P_\phi(s_f \mid G'), \qquad (4)$$

where $P_B(G \mid G')$ is a fixed distribution over the parent states of $G'$ (e.g., uniform distribution over parents). Deleu et al. (2022) showed that since all the states are complete here, satisfying the conditions (4) for all $G \to G'$ still induces a distribution over DAGs $\propto R(G)$. Therefore, to approximate the posterior distribution $P(G \mid \mathcal{D})$, they used $R(G) = P(\mathcal{D} \mid G)P(G)$ as the reward of $G$. In particular, this requires evaluating the marginal likelihood $P(\mathcal{D} \mid G)$ efficiently, which is feasible only for limited classes of models (e.g., linear Gaussian; Geiger and Heckerman, 1994).

# 3 Joint Bayesian inference of structure and parameters

Although Generative Flow Networks have been primarily applied to model distributions over discrete objects such as DAGs, Lahlou et al. (2023) showed that similar ideas could also be applied to continuous objects, and discrete-continuous hybrids. Building on top of DAG-GFlowNet, we propose here to approximate the joint posterior $P(G, \theta \mid \mathcal{D})$ over both the structure of the Bayesian Network $G$, but also the parameters of its conditional probability distributions $\theta$. Unlike in VBG though (Nishikawa-Toomey et al., 2023), we use a single GFlowNet to approximate this joint posterior. We call this model *JSP-GFN*, for Joint Structure and Parameters Bayesian inference with a GFlowNet.

## 3.1 Structure of the GFlowNet

Unlike in DAG-GFlowNet, where we model a distribution only over DAGs, here we need to define a GFlowNet whose complete states are pairs $(G, \theta)$, where $G$ is a DAG and $\theta$ is a set of (continuous-valued) parameters whose dimension may depend on $G$. Complete states are obtained through two phases (Figure 1): the DAG $G$ is first constructed one edge at a time, following Deleu et al. (2022), and then the corresponding parameters $\theta$ are generated, conditioned on $G$. We denote by $(G, \cdot)$ states where the DAG $G$ has no parameters $\theta$ associated to it (states in blue in Figure 1); they are intermediate states during the first phase of the GFlowNet, and do not correspond to valid samples of the induced distribution. Using the notations of Section 2.2, $(G, \theta) \in \mathcal{X}$, whereas $(G, \cdot) \in \mathcal{S} \backslash \mathcal{X}$.

Starting at the empty graph $(G_0, \cdot)$, the DAG is constructed one edge at a time during the first phase, following the forward transition probabilities $P_\phi(G' \mid G)$. This first phase ends when a special "stop" action is selected with $P_\phi$, indicating that we stop adding edges to the graph; the role of this "stop" action is detailed in Section 3.4. Then during the second phase, we generate $\theta$ conditioned on $G$, following the forward transition probabilities $P_\phi(\theta \mid G)$.[2] All the complete states $(G, \theta)$, for a fixed graph $G$ and any set of parameters $\theta$, can be seen as forming an (infinitely wide) tree rooted at $(G, \cdot)$.

Since we want this GFlowNet to approximate the joint posterior $P(G, \theta \mid \mathcal{D}) \propto P(\mathcal{D}, \theta, G)$, it is natural to define the reward function of a complete state $(G, \theta)$ as

$$R(G, \theta) = P(\mathcal{D} \mid \theta, G)P(\theta \mid G)P(G), \qquad (5)$$

where the likelihood model $P(\mathcal{D} \mid \theta, G)$ may be arbitrary (e.g., a neural network), and decomposes according to (1), $P(\theta \mid G)$ is the prior over parameters, and $P(G)$ the prior over graphs. When the dataset $\mathcal{D}$ is large, we can use a mini-batch approximation to the reward (see Section 3.5).

---

[2]Since the states of the GFlowNet here are pairs of objects, $P_\phi(\theta \mid G)$ (resp. $P_\phi(G' \mid G)$) is an abuse of notation, and represents $P_\phi((G, \theta) \mid (G, \cdot))$ (resp. $P_\phi((G', \cdot) \mid (G, \cdot))$).

## 3.2 Subtrajectory Balance conditions

To obtain a generative process that samples pairs of $(G, \theta)$ proportionally to the reward, the GFlowNet needs to satisfy some conditions such as (4). However, we saw in Section 2.3 that satisfying this particular formulation of the detailed balance conditions in (4) yields a distribution $\propto R(\cdot)$ only if all the states are complete; unfortunately, this is not the case here since there exists states of the form $(G, \cdot)$ corresponding to graphs without their associated parameters. Instead, we use a generalization of detailed balance to undirected paths of arbitrary length, called the *Subtrajectory Balance* conditions (SubTB; Malkin et al., 2022); we give a brief overview of SubTB in Appendix C.2.

More precisely, we consider SubTB for any undirected path of the form $(G, \theta) \leftarrow (G, \cdot) \rightarrow (G', \cdot) \rightarrow (G', \theta')$ in the GFlowNet (see Figure C.2), where $G'$ is therefore the result of adding a single edge to $G$. Since both ends of these undirected paths of length 3 are complete states, we show in Appendix C.3.1 that the SubTB conditions corresponding to undirected paths of this form can be written as

$$R(G', \theta')P_B(G \mid G')P_\phi(\theta \mid G) = R(G, \theta)P_\phi(G' \mid G)P_\phi(\theta' \mid G'). \tag{6}$$

Note that the SubTB condition above is very similar to the detailed balance condition in (4) used in DAG-GFlowNet, where the probability of terminating $P_\phi(s_f \mid G)$ has been replaced by $P_\phi(\theta \mid G)$, in addition to the reward now depending on both $G$ and $\theta$. Moreover, while it does not seem to appear in (6), the terminal state $s_f$ of the GFlowNet is still present implicitly, since we are forced to terminate once we have reached a complete state $(G, \theta)$, and therefore $P_\phi(s_f \mid G, \theta) = 1$ (see App. C.3.1).

Although there is no guarantee in general that satisfying the SubTB conditions would yield a distribution proportional to the reward, unlike with the detailed balance conditions (Bengio et al., 2023), the following theorem shows that the GFlowNet does induce a distribution $\propto R(G, \theta)$ if the SubTB conditions in (6) are satisfied for all pairs $(G, \theta)$ and $(G', \theta')$.

**Theorem 3.1.** *If the SubTB conditions in (6) are satisfied for all undirected paths of length 3 between any $(G, \theta)$ and $(G', \theta')$ of the form $(G, \theta) \leftarrow (G, \cdot) \rightarrow (G', \cdot) \rightarrow (G', \theta')$, then we have*

$$P_\phi^\top(G, \theta) \triangleq P_\phi(G \mid G_0)P_\phi(\theta \mid G) \propto R(G, \theta),$$

*where $P_\phi(G \mid G_0)$ is the marginal probability of reaching $G$ from the initial state $G_0$ with any (complete) trajectory $\tau = (G_0, G_1, \ldots, G_{T-1}, G)$:*

$$P_\phi(G \mid G_0) \triangleq \sum_{\tau : G_0 \rightsquigarrow G} \prod_{t=0}^{T-1} P_\phi(G_{t+1} \mid G_t),$$

*using the conventions $G_T = G$, and $P_\phi(G_0 \mid G_0) = 1$.*

The proof of this theorem is available in Appendix C.3.3. The marginal distribution $P_\phi^\top(G, \theta)$ is also called the *terminating state probability* in Bengio et al. (2023).

## 3.3 Learning objective

One way to find the parameters $\phi$ of the forward transition probabilities that enforce the SubTB conditions in (6) for all $(G, \theta)$ and $(G', \theta')$ is to transform this condition into a learning objective. For example, we could minimize a non-linear least squares objective (Bengio et al., 2021, 2023) of the form $\mathcal{L}(\phi) = \mathbb{E}_\pi[\Delta^2(\phi)]$, where the residuals $\Delta(\phi)$ depend on the conditions in (6), and $\pi$ is an arbitrary sampling distribution of complete states $(G, \theta)$ and $(G', \theta')$, with full support; see Malkin et al. (2023) for a discussion of the effect of $\pi$ on training GFlowNets, and Appendix D.1 for further details in the context of JSP-GFN.

In addition to the SubTB conditions for undirected paths of length 3 given in Section 3.2, we can also derive similar SubTB conditions for other undirected paths, for example those of the form $(G, \theta) \leftarrow (G, \cdot) \rightarrow (G, \tilde{\theta})$, where $\theta$ and $\tilde{\theta}$ are two possible sets of parameters associated with the same graph $G$ (see also Figure C.2). When the reward function $R(G, \theta)$ is differentiable wrt. $\theta$, which is the case here, we show in Appendix C.3.2 that we can write the SubTB conditions for these undirected paths of length 2 in differential form as

$$\nabla_\theta \log P_\phi(\theta \mid G) = \nabla_\theta \log R(G, \theta). \tag{7}$$

This condition is equivalent to the notion of score matching (Hyvärinen, 2005) to model unnormalized distributions, since $R(G, \theta)$ here corresponds to the unnormalized posterior distribution $P(\theta \mid G, \mathcal{D})$, where the normalization is over $\theta$. We also show in Appendix C.3.2 that incorporating this information about undirected paths of length 2, via the identity in (7), amounts to *preventing* backpropagation through $\theta$ and $\theta'$ (e.g., backpropagation with the reparametrization trick) in the objective

$$\mathcal{L}(\phi) = \mathbb{E}_\pi\left[\left(\log \frac{R\big(G', \perp(\theta')\big) P_B(G \mid G') P_\phi\big(\perp(\theta) \mid G\big)}{R\big(G, \perp(\theta)\big) P_\phi(G' \mid G) P_\phi\big(\perp(\theta') \mid G'\big)}\right)^2\right], \tag{8}$$

where $\perp$ denotes the "stop-gradient" operation. This is aligned with the recommendations of Lahlou et al. (2023) to avoid backpropagation through the reward in continuous GFlowNets. The pseudo-code for training JSP-GFN is available in Algorithm 1.

## 3.4 Parametrization of the forward transition probabilities

In Section 3.1, we saw that the process of generating $(G, \theta)$ follows two phases: first we construct $G$ one edge at a time, until we sample a specific "stop" action, at which point we sample $\theta$, conditioned on $G$. All these actions are sampled using the forward transition probabilities $P_\phi(G' \mid G)$ during the first phase, and $P_\phi(\theta \mid G)$ during the second one. Following Deleu et al. (2022), we parametrize these forward transition probabilities using a hierarchical model: we first decide whether we want to stop the first phase or not, with probability $P_\phi(\text{stop} \mid G)$; then, conditioned on this first decision, we either continue adding an edge to $G$ to reach $G'$ with probability $P_\phi(G' \mid G, \neg\text{stop})$ (phase 1), or sample $\theta$ with probability $P_\phi(\theta \mid G, \text{stop})$ (phase 2). This hierarchical model can be written as

$$P_\phi(G' \mid G) = \big(1 - P_\phi(\text{stop} \mid G)\big) P_\phi(G' \mid G, \neg\text{stop}) \tag{9}$$

$$P_\phi(\theta \mid G) = P_\phi(\text{stop} \mid G) P_\phi(\theta \mid G, \text{stop}). \tag{10}$$

We use neural networks to parametrize each of the three components necessary to define the forward transition probabilities. Unlike in DAG-GFlowNet though, which uses a linear Transformer to define $P_\phi(\text{stop} \mid G)$ and $P_\phi(G' \mid G, \neg\text{stop})$, we use a combination of graph network (Battaglia et al., 2018) and self-attention blocks (Vaswani et al., 2017) to encode information about the graph $G$, which appears in the conditioning of all the quantities of interest. This common backbone returns a graph embedding $\boldsymbol{g}$ of $G$, as well as 3 embeddings $\boldsymbol{u}_i, \boldsymbol{v}_i, \boldsymbol{w}_i$ for each node $X_i$ in $G$

$$\boldsymbol{g}, \{\boldsymbol{u}_i, \boldsymbol{v}_i, \boldsymbol{w}_i\}_{i=1}^d = \text{SelfAttention}_\phi\big(\text{GraphNet}_\phi(G)\big).$$

We can parametrize the probability of selecting the "stop" action using $\boldsymbol{g}$ with $P_\phi(\text{stop} \mid G) = f_\phi(\boldsymbol{g})$, where $f_\phi$ is a neural network with a sigmoid output; note that if we can't add any edge to $G$ without creating a cycle, we force the end of the first phase by setting $P_\phi(\text{stop} \mid G) = 1$. Inspired by Lorch et al. (2021), the probability of moving from $G$ to $G'$ by adding the edge $X_i \to X_j$ is parametrized by

$$P_\phi(G' \mid G, \neg\text{stop}) \propto \boldsymbol{m}_{ij} \exp\big(\boldsymbol{u}_i^\top \boldsymbol{v}_j\big), \tag{11}$$

where $\boldsymbol{m}_{ij}$ is a binary mask indicating whether adding $X_i \to X_j$ is a valid action (i.e., if it is not already present in $G$, and if it doesn't introduce a cycle; Deleu et al., 2022). Finally, the probability of selecting the parameters $\theta_i$ of the CPD for the variable $X_i$ is parametrized with a multivariate Normal distribution with diagonal covariance (unless specified otherwise)

$$P_\phi(\theta_i \mid G, \text{stop}) = \mathcal{N}\big(\theta_i \mid \boldsymbol{\mu}_\phi(\boldsymbol{w}_i), \boldsymbol{\sigma}_\phi^2(\boldsymbol{w}_i)\big), \tag{12}$$

where $\boldsymbol{\mu}_\phi$ and $\boldsymbol{\sigma}_\phi^2$ are two neural networks, with appropriate non-linearities to guarantee that $\boldsymbol{\sigma}_\phi^2(\boldsymbol{w}_i)$ is a well-defined diagonal covariance matrix. Note that $P_\phi(\theta_i \mid G, \text{stop})$ effectively approximates the posterior distribution $P(\theta_i \mid G, \mathcal{D})$ once fully trained. Moreover, in addition to being an approximation of the joint posterior $P(G, \theta \mid \mathcal{D})$, the GFlowNet also provides an approximation of the marginal posterior $P(G \mid \mathcal{D})$, by only following the first phase of the generation process (to generate $G$) until the "stop" action is selected, and not continuing into the generation of $\theta$.

## 3.5 Mini-batch training

Throughout the paper, we have assumed that we had access to the full dataset of observations $\mathcal{D}$ in order to compute the reward $R(G, \theta)$ in (5). However, beyond the capacity to have an arbitrary likelihood model $P(\mathcal{D} \mid \theta, G)$ (e.g., non-linear), another advantage of approximating the joint posterior

$P(G, \theta \mid \mathcal{D})$ is that we can train the GFlowNet using mini-batches of observations. Concretely, for a mini-batch $\mathcal{B}$ of $M$ observations sampled uniformly at random from the dataset $\mathcal{D}$, we can define

$$\log \widehat{R}_{\mathcal{B}}(G, \theta) = \log P(\theta \mid G) + \log P(G) + \frac{N}{M} \sum_{\boldsymbol{x}^{(m)} \in \mathcal{B}} \log P(\boldsymbol{x}^{(m)} \mid G, \theta), \qquad (13)$$

which is an unbiased estimate of the log-reward. The following proposition shows that minimizing the estimated loss based on (13) wrt. the parameters $\phi$ of the GFlowNet also minimizes the original objective in Section 3.3.

**Proposition 3.2.** *Suppose that $\mathcal{B}$ is a mini-batch of $M$ observations sampled uniformly at random from the dataset $\mathcal{D}$, and let $\widehat{\mathcal{L}}_{\mathcal{B}}(\phi)$ be the learning objective defined in Section 3.3, where the reward has been replaced by the estimate $\widehat{R}_B(G, \theta)$ in (13). Then we have $\mathcal{L}(\phi) \leq \mathbb{E}_{\mathcal{B}}\big[\widehat{\mathcal{L}}_{\mathcal{B}}(\phi)\big]$.*

The proof is available in Appendix C.3.4, only relies on the convexity of the square function to conclude, but no other property of this function; in practice, we use the Huber loss instead of the square loss for stability, which is also a convex function. In the case of the square loss, Proposition C.1 gives a stronger result in terms of unbiasedness of the gradient estimator.

## 4 Related work

**Bayesian Structure Learning.** There is a vast literature applying Markov chain Monte Carlo (MCMC) methods to approximate the marginal posterior $P(G \mid \mathcal{D})$ over the graphical structures of Bayesian Networks (Madigan et al., 1995; Friedman and Koller, 2003; Giudici and Castelo, 2003; Viinikka et al., 2020). However, since the parameter space in which $\theta$ lives depends on the graph structure $G$, approximating the joint posterior $P(G, \theta \mid \mathcal{D})$ using MCMC requires additional trans-dimensional updates (Fronk, 2002), and has therefore received less attention than the marginal case.

Variational methods have been proposed to approximate the marginal posterior too (Annadani et al., 2021; Charpentier et al., 2022). Similar to MCMC though, approximating the joint posterior has also been less studied than its marginal counterpart, with the notable exceptions of DiBS (Lorch et al., 2021) and BCD Nets (Cundy et al., 2021). We provide an extensive qualitative comparison between our method JSP-GFN and prior variational inference and GFlowNet methods in Appendix A.

**Generative Flow Networks.** While they were initially developed to encourage the discovery of diverse molecules (Bengio et al., 2021), GFlowNets proved to be a more general framework to describe distributions over composite objects that can be constructed sequentially (Bengio et al., 2023). The objective of the GFlowNet is to enforce a conservation law such as the flow-matching conditions in (2), indicating that the total amount of flow going into any state is equal to the total outgoing flow, with some residual given by the reward. Alternative conditions, sometimes bypassing the need to work with flows altogether, have been proposed in order to learn these models more efficiently (Malkin et al., 2022; Madan et al., 2022; Pan et al., 2023). By amortizing inference, and thus treating it as an optimization problem, GFlowNets find themselves deeply rooted in the variational inference literature (Malkin et al., 2023; Zimmermann et al., 2022), and are connected to other classes of generative models (Zhang et al., 2022). Beyond Bayesian structure learning (Deleu et al., 2022), GFlowNets have also applications in modeling Bayesian posteriors for variational EM (Hu et al., 2023), combinatorial optimization (Zhang et al., 2023), biological sequence design (Jain et al., 2022), as well as scientific discovery at large (Jain et al., 2023).

Closely related to our work, Nishikawa-Toomey et al. (2023) proposed to learn the joint posterior $P(G, \theta \mid \mathcal{D})$ over structures and parameters with a GFlowNet, combined with Variational Bayes to circumvent the challenge of learning a distribution over continuous quantities $\theta$ with a GFlowNet. Atanackovic et al. (2023) also used a GFlowNet called *DynGFN* to approximate the posterior of a dynamical system. Similar to (Nishikawa-Toomey et al., 2023) though, they used the GFlowNet only to approximate the distribution over graphs $G$, making the parameters $\theta$ a deterministic function of $G$ (i.e., $P(\theta \mid G, \mathcal{D}) \approx \delta(\theta \mid G; \phi)$). Here, we leverage the recent advances extending these models to general sample spaces (Lahlou et al., 2023), including continuous spaces (Li et al., 2023), in order to model the joint posterior within a single GFlowNet.

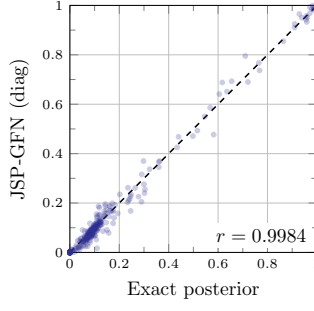

| | Edge features | | $\mathbb{E}_{G,\theta}\big[-\log P(\theta \mid G, \mathcal{D})\big]$ |
|---|---|---|---|
| | RMSE | Pearson's $r$ | |
| MH-MC[3] | $0.357 \pm 0.022$ | $0.067 \pm 0.143$ | $5.39 \pm 1.41 \times 10^2$ |
| Gibbs-MC[3] | $0.357 \pm 0.022$ | $0.028 \pm 0.127$ | $9.02 \pm 1.54 \times 10^5$ |
| B-GES[*] | $0.263 \pm 0.070$ | $0.635 \pm 0.180$ | $1.56 \pm 0.97 \times 10^2$ |
| B-PC[*] | $0.305 \pm 0.057$ | $0.570 \pm 0.138$ | $1.57 \pm 0.87 \times 10^2$ |
| DiBS | $0.312 \pm 0.038$ | $0.737 \pm 0.071$ | $9.49 \pm 7.34 \times 10^3$ |
| BCD Nets | $0.215 \pm 0.055$ | $0.819 \pm 0.097$ | $7.04 \pm 3.21 \times 10^1$ |
| VBG | $0.237 \pm 0.037$ | $0.816 \pm 0.064$ | $1.24 \pm 0.49 \times 10^2$ |
| JSP-GFN (diag) | $\mathbf{0.018 \pm 0.005}$ | $\mathbf{0.998 \pm 0.001}$ | $\mathbf{-4.91 \pm 0.51 \times 10^0}$ |
| JSP-GFN (full) | $\mathbf{0.019 \pm 0.007}$ | $\mathbf{0.998 \pm 0.001}$ | $\mathbf{-5.00 \pm 0.52 \times 10^0}$ |

(a) Edge features

(b) Quantitative comparison with the exact posterior

Figure 2: Comparison with the exact posterior distribution, on small graphs with $d = 5$ nodes. (a) Comparison of the edge features computed with the exact posterior (x-axis) and the approximation given by JSP-GFN (y-axis); each point corresponds to an edge $X_i \to X_j$ for each of the 20 datasets. (b) Quantitative evaluation of different methods for joint posterior approximation, both in terms of edge features and cross-entropy of sampling distribution and true posterior $P(\theta \mid G, \mathcal{D})$; all values correspond to the mean and $95\%$ confidence interval across the 20 experiments.

## 5 Experimental results

### 5.1 Joint posterior over small graphs

We can evaluate the accuracy of the approximation returned by JSP-GFN by comparing it with the exact joint posterior distribution $P(G, \theta \mid \mathcal{D})$. Computing this exact posterior can only be done in limited cases only, where (1) $P(\theta \mid G, \mathcal{D})$ can be computed analytically, and (2) for a small enough $d$ such that all the DAGs can be enumerated in order to compute $P(G \mid \mathcal{D})$. We consider here models over $d = 5$ variables, with linear Gaussian CPDs. We generate 20 different datasets of $N = 100$ observations from randomly generated Bayesian Networks. Details about data generation and modeling are available in Appendix D.2.

The quality of the joint posterior approximations is evaluated separately for $G$ and $\theta$. For the graphs, we compare the approximation and the exact posterior on different marginals of interest, also called *features* (Friedman and Koller, 2003); e.g., the *edge feature* corresponds to the marginal probability of a specific edge being in the graph. Figure 2 (a) shows a comparison between the edge features computed with the exact posterior and with JSP-GFN, proving that it can accurately approximate the edge features of the exact posterior, despite the modeling bias discussed in Appendix D.2.2. Compared to other methods in Figure 2 (b), JSP-GFN offers significantly more accurate approximations of the posterior, at least relative to the edge features. This observation still holds on the path and Markov features (Deleu et al., 2022); see Appendix D.2.2.

To evaluate the performance of the different methods as an approximation of the posterior over $\theta$, we also estimate the cross-entropy between the sampling distribution of $\theta$ given $G$ and the exact posterior $P(\theta \mid G, \mathcal{D})$. This measure will be minimized if the model correctly samples parameters from the true $P(\theta \mid G, \mathcal{D})$; details about this metric are given in Appendix D.2.3. In Figure 2 (b), we observe that again both versions of JSP-GFN sample parameters $\theta$ that are significantly more probable under the exact posterior compared to other methods.

### 5.2 Gaussian Bayesian Networks from simulated data

To evaluate whether our observations hold on larger graphs, we also evaluated the performance of JSP-GFN on data simulated from larger Gaussian Bayesian Networks, with $d = 20$ variables. In addition to linear CPDs, as in Section 5.1, we experimented with non-linear Gaussian Bayesian Networks, where the CPDs are parametrized by neural networks. Following Lorch et al. (2021), we parametrized the CPDs of each variable with a 2-layer MLP, for a total of $|\theta| = 2,220$ parameters. For both experimental settings, we used datasets of $N = 100$ observations simulated from (randomly generated) Bayesian Networks; additional experimental details are provided in Appendix D.3.

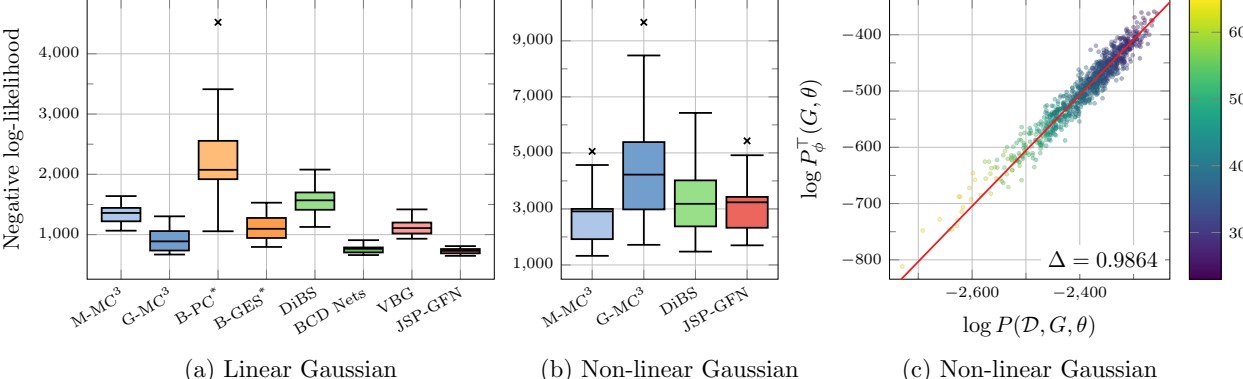

(a) Linear Gaussian        (b) Non-linear Gaussian        (c) Non-linear Gaussian

Figure 3: Evaluation of JSP-GFN on Gaussian Bayesian Networks. (a-b) Comparison of the negative log-likelihood (NLL) on $N' = 100$ held-out observations for different Bayesian structure learning methods, aggregated across 20 experiments on different datasets $\mathcal{D}$. (c) Linear correlation between the log-reward (x-axis) and the terminating state log-probability (y-axis) for $1,000$ samples $(G, \theta)$ from JSP-GFN; the color of each point indicates the number of edges in the corresponding graph. $\Delta$ represents the slope of a linear function fitted using RANSAC (Fischler and Bolles, 1981).

We compared JSP-GFN against two methods based on MCMC (MH-MC³ & Gibbs-MC³; Madigan et al., 1995) and DiBS (Lorch et al., 2021) on both experiments, as well as two bootstrapping algorithms (B-GES* & B-PC*; Friedman et al., 1999), BCD Nets (Cundy et al., 2021) and VBG (Nishikawa-Toomey et al., 2023) for the experiment with linear Gaussian CPDs, as they are not applicable for non-linear CPDs. In Figure 3 (a-b), we report the performance of these joint posterior approximations in terms of the (expected) negative log-likelihood (NLL) on held-out observations. We observe that JSP-GFN achieves a lower NLL than any other method on linear Gaussian models and is competitive on non-linear Gaussian models.

We chose the NLL over some other metrics, typically comparing with the ground-truth graphs used for data generation, since it is more representative of the performance of these methods on downstream tasks (i.e., predictions on unseen data), and measures the quality of the joint posterior instead of only the marginal over graphs. This choice is aligned with the shortcomings of these other metrics highlighted by Lorch et al. (2022), and it is further justified in Appendix D.3.3. Nevertheless, we also report the expected Structural Hamming Distance (SHD), as well as the area under the ROC curve (AUROC) in Appendix D.3.3 for completeness. To complement these metrics, and in order to assess the quality of the approximation of the posterior in the absence of reference $P(G, \theta \mid \mathcal{D})$, we also show in Figure 3 (c) how the terminating state log-probability $\log P_\phi^\top(G, \theta)$ of JSP-GFN correlates with the log-reward for a non-linear Gaussian model. Indeed, as stated in Theorem 3.1, we should ideally have $\log P_\phi^\top(G, \theta)$ perfectly correlated with $\log R(G, \theta)$ with slope 1, as

$$\log P_\phi^\top(G, \theta) \approx \log P(G, \theta \mid \mathcal{D}) = \log R(G, \theta) - \log P(\mathcal{D}). \tag{14}$$

We can see that there is indeed a strong linear correlation across multiple samples $(G, \theta)$ from JSP-GFN, with a slope $\Delta$ close to 1, suggesting that the GFlowNet is again an accurate approximation of the joint posterior, *at least* around the modes it captures. Details about how $\log P_\phi^\top(G, \theta)$ is estimated are available in Appendix D.3.2.

## 5.3   Learning biological structures from real data

We finally evaluated JSP-GFN on real-world biological data for two separate tasks: the discovery of protein signaling networks from flow cytometry data (Sachs et al., 2005), as well as the discovery of a small gene regulatory network from gene expression data. The flow cytometry dataset consists of $N = 4,200$ measurements of $d = 11$ phosphoproteins from 7 different experiments, meaning that this dataset contains a mixture of both observational and interventional data. Furthermore, this dataset has been discretized into 3 states, representing the level of activity (Eaton and Murphy, 2007). For the gene expression dataset, we used a subset of $N = 2,628$ observations of $d = 61$ genes from (Sethuraman et al., 2023). Details about the experimental setups are available in Appendix D.4.

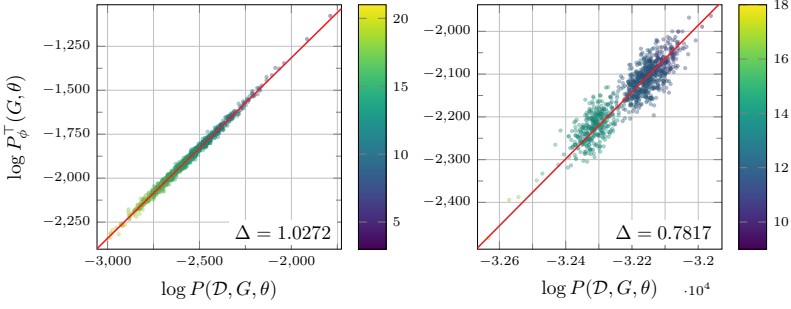

(a) Flow cytometry data (subsample, $N = 100$)

(b) Flow cytometry data

(c) Negative log-likelihood

|  | Flow cytometry $d = 11, N = 4,200$ | Gene expression $d = 61, N = 2,628$ |
| --- | --- | --- |
| MH-MC$^3$ | $\mathbf{2.578 \times 10^4}$ | $1.073 \times 10^6$ |
| Gibbs-MC$^3$ | $3.490 \times 10^5$ | $4.882 \times 10^6$ |
| JSP-GFN | $2.679 \times 10^4$ | $\mathbf{2.651 \times 10^5}$ |

Figure 4: Performance of JSP-GFN on real-world biological data. (a) Comparison of the terminating state log-probability $\log P_\phi^\top(G, \theta)$ returned by JSP-GFN with the log-reward $\log R(G, \theta)$ on a subsample of $N = 100$ datapoints of the flow cytometry dataset $\mathcal{D}$. (b) Same comparison with the full dataset $\mathcal{D}$ of size $N = 4,200$. (c) Comparison of JSP-GFN with methods based on MCMC on both flow cytometry data and gene expression data, in terms of negative (interventional) log-likelihood on held-out data.

At this scale, using the whole dataset $\mathcal{D}$ to evaluate the reward becomes impractical, especially for non-linear models. Fortunately, we showed in Section 3.5 that we can use an (unbiased) estimate of the reward, based on mini-batches of data, in place of $R(G, \theta)$ in the loss function (8). In both experiments, we used non-linear models, where all the CPDs are parametrized with a 2-layer MLP. Figure 4 shows similar correlation plots as Figure 3 (c), along with an evaluation of the NLL on unseen observations and interventions. Beyond the ability to work with real data, these experiments allow us to highlight some other capacities of JSP-GFN: (1) handling discrete and (2) interventional data (flow cytometry), as well as (3) learning a distribution over larger graphs (gene expression).

To measure the quality of the posterior approximation returned by JSP-GFN, we compare in Figure 4 the terminating state log-probability $\log P_\phi^\top(G, \theta)$ with the log-reward $\log R(G, \theta)$, similar to Figure 3. We observe that there is correlation between these two quantities; unlike in Figure 3 though, we observe that the slope is not close to 1, suggesting that JSP-GFN *underestimates* the probability of $(G, \theta)$. We also observe that the graphs are "clustered" together; this can be explained by the fact that the posterior approximation is concentrated at only a few graphs, since the size of the dataset $\mathcal{D}$ is larger. To confirm this observation, we show in Figure 4 (a) a similar plot on a subsample of $N = 100$ datapoints randomly sampled from $\mathcal{D}$, matching the experimental setting of Section 5.2. In this case, we observe a much closer linear fit, with a slope closer to 1.

In addition to the comparison to the log-reward, we also compare in Figure 4 (c) JSP-GFN with 2 methods based on MCMC in terms of their negative log-likelihood on held-out data. We can see that JSP-GFN is competitive, and even out-performs MCMC on the more challenging problem of the discovery of gene regulatory networks from gene expression data, where the dimensionality of the problem is much larger ($d = 61$). Note that the values reported for the discovery of protein signaling networks from flow cytometry data correspond to the negative *interventional* log-likelihood, on interventions unseen in $\mathcal{D}$.

## 6 Conclusion

We have presented JSP-GFN, an approach to approximate the joint posterior distribution over the structure of a Bayesian Network along with its parameters using a single GFlowNet. We have shown that our method faithfully approximates the joint posterior on both simulated and real data, and compares favorably against existing Bayesian structure learning methods. In line with Appendix B, future work should consider using more expressive distributions, such as those parametrized by normalizing flows or diffusion processes, to approximate the posteriors over continuous parameters, which would enable Bayesian inference over parameters in more complex generative models.

## Acknowledgements

We would like to thank António Góis, Salem Lahlou, Romain Lopez, Cristian Dragos Manta, and Mansi Rankawat for the useful discussions about the project and their feedback on the paper. We would like to also thank Lars Lorch for releasing the code for DiBS (Lorch et al., 2021), along with useful baselines used in this paper, as well as Chris Cundy for releasing the code for BCD Nets and for his help with the code. This research was enabled by compute resources provided by Mila (mila.quebec).

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

# Appendix

## A  Positioning JSP-GFN in the Bayesian structure learning literature

We give a comparison between our method JSP-GFN, and various methods based on variational inference in Table A.1. We also include methods based on GFlowNets, namely DAG-GFlowNet (Deleu et al., 2022) and VBG (Nishikawa-Toomey et al., 2023), as they are effectively variational methods (Malkin et al., 2023; Zimmermann et al., 2022).

Table A.1: Comparison of different methods based on variational inference and GFlowNets for Bayesian structure learning. See the text for a detailed description of each category.

| | Joint $G$ & $\theta$ | Non Linear | DAG Support | Discrete Obs. | Max. Parents | Sampler | Mini Batch |
|---|---|---|---|---|---|---|---|
| VCN (Annadani et al., 2021) | ✗ | ✗ | ✗ | ● | ✗ | ✓ | ✗ |
| BCD Nets (Cundy et al., 2021) | ✓ | ✗ | ✓ | ✗ | ✗ | ✓ | ✗ |
| DiBS (Lorch et al., 2021) | ✓ | ✓ | ✗ | ✓ | ✗ | ✗ | ✓ |
| TRUST (Wang et al., 2022) | ✗ | ✗ | ✓ | ● | ✗ | ● | ✗ |
| VI-DP-DAG (Charpentier et al., 2022) | ✗ | ✓ | ✓ | ✗ | ✗ | ✓ | ✗ |
| AVICI (Lorch et al., 2022) | ✗ | ✓ | ✗ | ● | ✗ | ✓ | ✗ |
| DAG-GFlowNet (Deleu et al., 2022) | ✗ | ✗ | ✓ | ✓ | ✓ | ✓ | ✗ |
| VBG (Nishikawa-Toomey et al., 2023) | ✓ | ● | ✓ | ● | ✓ | ✓ | ✗ |
| **JSP-GFN (Ours)** | ✓ | ✓ | ✓ | ✓ | ✓ | ✓ | ✓ |

**Joint $G$ & $\theta$.** This category indicates whether the model can approximate the joint posterior distribution $P(G, \theta \mid \mathcal{D})$ over both graphical structures $G$ and parameters of the CPDs $\theta$, or if they are limited to approximating the marginal posterior $P(G \mid \mathcal{D})$. As we have seen in Section 1, approximations of the marginal posteriors limit the classes of models these methods can be applied to, namely those where the marginal likelihood can be computed analytically.

**Non-Linear.** This indicates whether the model can be applied to Bayesian Networks whose CPDs are parametrized by a non-linear function (e.g., a neural network). While most methods approximating the marginal distribution may be applied to non-linear CPDs parametrized by a Gaussian Process (von Kügelgen et al., 2019), we only consider here methods that explicitly handle non-linearity (e.g., this eliminates DAG-GFlowNet (Deleu et al., 2022), since the authors only considered a linear Gaussian and discrete settings). Annadani et al. (2021) mentioned the extension of VCN to non-linear causal models as future work. While VBG (Nishikawa-Toomey et al., 2023) has only been applied to linear Gaussian models, the framework may also be applicable to non-linear models.

**DAG Support.** This indicates whether the posterior approximation is guaranteed to have support over the space of DAGs. VCN (Annadani et al., 2021) and DiBS (Lorch et al., 2021) only encourage acyclicity via a prior term, inspired by continuous relaxations of the acyclicity constraint (Zheng et al., 2018), meaning the those methods may return graphs containing cycles; for example in practice, Deleu et al. (2022) reports that $1.50\%$ of the graphs returned by DiBS contain cycles (for $d = 11$). AVICI (Lorch et al., 2022) uses a similar prior term when applied to Structural Causal Models (SCMs), although in general this framework does not enforce acyclicity by design, to allow flexibility on other domains (e.g., for modeling gene regulatory networks; Sethuraman et al., 2023). TRUST (Wang et al., 2022) guarantees acyclicity via a distribution over variable orders, that can be learned using Sum-Product Networks.

**Discrete Observations.** This indicates whether the posterior approximation may be applied to Bayesian Networks with discrete random variables. Although VCN (Annadani et al., 2021) was only applied to linear Gaussian models, the authors mention that this approach is also applicable to discrete random variables. Similarly, while there is no experiment in (Lorch et al., 2021) applying DiBS to a discrete domain, this extension can be found in the official code released. AVICI (Lorch et al., 2022) assumes access to a generative model $P(\mathcal{D} \mid G)$, making it possibly applicable to discrete domains as

well. Since it builds on DAG-GFlowNet (Deleu et al., 2022), VBG (Nishikawa-Toomey et al., 2023) should also inherit its properties, and therefore may also be applicable to discrete random variables. Similarly, since TRUST (Wang et al., 2022) may use DiBS as its underlying routine for structure learning, it should also inherit the properties of DiBS.

**Maximum Parents.**   This category indicates whether a maximum number of parents can be specified for each variable in the DAGs returned by each method. Although this is a very common constraint used in the structure learning literature to improve efficiency (Koller and Friedman, 2009), none of the variational methods for Bayesian structure learning allow for such a (hard) constraint. Some methods may introduce a sparsity-inducing prior (Lorch et al., 2021; Cundy et al., 2021), or use post-processing of the sampled DAGs (Charpentier et al., 2022) to reduce the number of edges in the sampled graphs. This can be naturally added in a GFlowNet, by masking out the actions adding certain edges that would violate this constraint; in fact, in the official code released for both DAG-GFlowNet (Deleu et al., 2022) and VBG (Nishikawa-Toomey et al., 2023) (both using the same environment), this option is available.

**Sampler.**   This category indicates whether one can sample graphs and parameters from the model once fully trained. DiBS (Lorch et al., 2021) uses a particle-based approach (Liu and Wang, 2016) to approximate the posterior (marginal, or joint), and therefore the number of particles is fixed ahead of time; once fully trained, it is impossible to sample new pairs of graphs and parameters from this model. TRUST (Wang et al., 2022) can also use Gadget (an MCMC approach to Bayesian structure learning; Viinikka et al., 2020) as its routine for structure learning, and therefore this would allow sampling from the trained model.

**Mini-Batch.**   This indicates whether the model can be updated with mini-batch of observations from $\mathcal{D}$, or if the full dataset must be used. DiBS (Lorch et al., 2021) uses mini-batch updates for their experiments on protein signaling networks, where the number of datapoints $N = 7466$ is large. Note that unlike JSP-GFN here, neither DAG-GFlowNet (Deleu et al., 2022) nor VBG (Nishikawa-Toomey et al., 2023) may be updated using mini-batches, since the maginalization over $\theta$ makes all observations in $\mathcal{D}$ mutually *dependent* (conditioned on $G$). See Appendix C.3.4 for details on how to use mini-batch training with JSP-GFN.

# B   Broader impact & limitations

## B.1   Broader impact

While structure learning of Bayesian Networks constitutes one of the foundations of *causal discovery* (also known as causal structure learning), it is important to emphasize that shy of any assumptions, the relationships learned from observations in a Bayesian Network are in general *not* causal, but merely statistical associations. As such, care must be taken interpreting the graphs sampled with JSP-GFN (or any other Bayesian structure learning method considered in this paper) as being causal. This is especially true when applying structure learning methods to the problem of scientific discovery (Jain et al., 2023). Assumptions that would allow causal interpretation of the graphs include using interventional data (as in Section 5.3), or parametric assumptions.

Although the graphs returned by JSP-GFN are not guaranteed to be causal, treating structure learning from a Bayesian perspective allows us to view identification of the causal relationships in a *softer* way. Indeed, instead of returning a single graph which could be harmful from a causal perspective (notably due to the lack of data, see also Section 1), having a posterior distribution over Bayesian Networks allows us to average out any possible model that can explain the data.

## B.2   Limitations

**Expressivity of the posterior approximation.**   Throughout this paper, we use Normal distributions (with a diagonal covariance, except for *JSP-GFN (full)* in Section 5.1) to parametrize the approximation of the posterior over parameters $P_\phi(\theta \mid G, \text{stop})$ (see Section 3.4). This limits its expressivity to unimodal distributions only, and is an assumption which is commonly used with Bayesian neural networks. However in general, the posterior distribution $P(\theta \mid G, \mathcal{D})$ may be highly multimodal, especially when the model is non-linear (the posterior is Normal when the model is linear Gaussian,

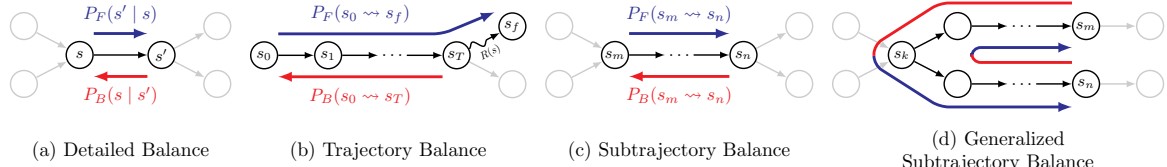

Figure C.1: Illustration of the different GFlowNet objectives. (a) The detailed balance condition operates at the level of transitions $s \to s'$, whereas (b) the trajectory balance condition operates on complete trajectories $s_0 \rightsquigarrow s_f$. (c) The subtrajectory balance condition operates on partial trajectories $s_m \rightsquigarrow s_n$, and can be (d) generalized to undirected paths with a common ancestor $s_k$. We use $P_F(s_m \rightsquigarrow s_n)$ to denote the product of $P_F$ along the path $s_m \rightsquigarrow s_n$ (and similarly for $P_B$).

see Appendix D.5.1). To see this, consider a non-linear model whose CPDs are parametrized by a 2-layer MLP (as in Sections 5.2 and 5.3). The weights and biases of both layers can be transformed in such a way that the hidden units get permuted, while preserving the outputs; in other words, there are many sets of parameters $\theta$ leading to the same likelihood function, and under mild assumptions on the priors $P(\theta \mid G)$ and $P(G)$, they would have the same posterior probability $P(\theta \mid G, \mathcal{D})$.

To address this issue of unimodality, we can use more expressive posterior approximations $P_\phi(\theta \mid G, \text{stop})$, such as ones parametrized with diffusion-based models, or with normalizing flows; both of these models are drop-in replacements in JSP-GFN, since their likelihood can be explicitly computed. An alternative is also to consider multiple steps of a continuous GFlowNet (Lahlou et al., 2023), instead of a single one, to generate $\theta$.

**Biological plausibility of the acyclicity assumption.** One of the strengths of JSP-GFN, and DAG-GFlowNet before it (Deleu et al., 2022), is the capacity to obtain a distribution over the DAG structure of a Bayesian Network (and its parameters). The acyclicity assumption is particularly important in order to properly define the likelihood model in (1). However in some domains, such as biological systems, there may exist some feedback processes that cannot be captured by acyclic graphs (Mooij et al., 2020). In particular, the DAGs found in Section 5.3 and Appendix D.4 must be carefully interpreted. As a general framework though, the GFlowNet used in JSP-GFN can be adapted to ignore the acyclic nature of the graphs sampled by ignoring parts of the mask $m$ in Sec. 3.4. Alternatively, we can view the generation of a cyclic graph by unrolling it, as in Atanackovic et al. (2023).

## C   Details about Generative Flow Networks

Throughout this section, we will use both $P_F$ and $P_\phi$ (to emphasize the parametrization on $\phi$, as in the main text) to denote equally the forward transition probability—the notation $P_F$ being more commonly used in the literature on GFlowNet (Bengio et al., 2023).

### C.1   Alternative conditions

GFlowNets were initially introduced using the flow-matching conditions (Bengio et al., 2021), as described in Section 2.2. However, there have been multiple alternative conditions that, once satisfied, also offer the same guarantees as the original flow-matching conditions (namely, a GFlowNet satisfying any of those conditions would sample complete states proportionally to the reward).

One of those alternative conditions are the *detailed balance* conditions (Bengio et al., 2023), inspired by the literature on Markov chains. These conditions are given for any transition $s \to s'$ in the GFlowNet as

$$F(s)P_F(s' \mid s) = F(s')P_B(s \mid s') \tag{C.1}$$

where $F(s)$ is a flow function, that may also be parametrized by a neural network. The detailed balance condition is illustrated in Figure C.1 (a). Bengio et al. (2023) showed that if the detailed balance conditions are satisfied for all the transitions $s \to s'$ in the GFlowNet, then the distribution induced by the GFlowNet is also proportional to $R(s)$. Deleu et al. (2022) adapted the detailed balance conditions in the case where all the states of the GFlowNet are complete, in order to avoid having to learn a separate flow function (see Section 2.3).

Another alternative condition called the *detailed balance* conditions (Malkin et al., 2022), operates not at the level of transitions, but at the level of complete trajectories. For a complete trajectory $\tau = (s_0, s_1, \ldots, s_T, s_f)$, the trajectory balance condition is given by

$$Z \prod_{t=1}^{T} P_F(s_{t+1} \mid s_t) = R(s_T) \prod_{t=1}^{T-1} P_B(s_t \mid s_{t+1}), \tag{C.2}$$

with the convention $s_{T+1} = s_f$, and where $Z$ is the partition function of the distribution (i.e., $Z = \sum_{x \in \mathcal{X}} R(x)$); in practice, $Z$ is a parameter of the model that is being learned alongside the forward and backward transition probabilities. The trajectory balance condition is illustrated in Figure C.1 (b). Again, if the trajectory balance conditions are satisfied for all complete trajectories in the GFlowNet, then the induced distribution is proportional to $R(s)$.

## C.2 Subtrajectory balance conditions

Also introduced in (Malkin et al., 2022), the *subtrajectory balance* conditions are a generalization of both the detailed balance and trajectory balance conditions to partial trajectories of arbitrary length. For a partial trajectory $\tau = (s_m, s_{m+1}, \ldots, s_n)$, the subtrajectory balance condition is given by

$$F(s_m) \prod_{t=m}^{n-1} P_F(s_{t+1} \mid s_t) = F(s_n) \prod_{t=m}^{n-1} P_B(s_t \mid s_{t+1}), \tag{C.3}$$

where again $F(s)$ is a flow function (as in (C.1)). This condition encompasses both conditions in Appendix C.1, since we can recover the detailed balance condition in (C.1) with partial trajectories of length 1 (i.e., transitions), and also the trajectory balance condition in (C.2) with complete trajectories (note that $F(s_0) = Z$; Bengio et al., 2023). The subtrajectory balance condition is illustrated in Figure C.1 (c). Madan et al. (2022) also proposed to combine subtrajectory balance conditions for partial trajectories of different lengths to create a novel objective called $\mathrm{SubTB}(\lambda)$, inspired by $\mathrm{TD}(\lambda)$ in the reinforcement learning literature.

This subtrajectory balance condition in (C.3) can also be generalized to undirected paths going "back and forth" (Malkin et al., 2022). For an undirected path between $s_m$ and $s_n$, this (generalized) subtrajectory balance condition can be written as

$$F(s_m) \prod_{t=k}^{m-1} P_B(s_t \mid s_{t+1}) \prod_{t=k}^{n-1} P_F(s_{t+1} \mid s_t) = F(s_n) \prod_{t=k}^{n-1} P_B(s_t \mid s_{t+1}) \prod_{t=k}^{m-1} P_F(s_{t+1} \mid s_t),$$
$$\tag{C.4}$$

where $s_k$ is a common ancestor of both $s_m$ and $s_n$. This condition is illustrated in Figure C.1 (d). While these subtrajectory balance conditions (generalized or not) offer more flexibility, they are guaranteed to yield a GFlowNet inducing a distribution proportional to $R(s)$ *only* if these conditions are satisfied for all the partial trajectories of any length. In particular, they provide no guarantee in general if those conditions are satisfied for all partial trajectories of fixed length, which is the case in this paper (see Section 3.2). Although this result may be extended with weaker assumptions, we prove in Appendix C.3.3 that the GFlowNet does induce a distribution $\propto R(s)$ in our case.

## C.3 Proofs

### C.3.1 Subtrajectory balance conditions for undirected paths of length 3

Consider an undirected path of length 3 of the form $(G, \theta) \leftarrow (G, \cdot) \rightarrow (G', \cdot) \rightarrow (G', \theta')$, where $G'$ is the result of adding a new edge to the DAG $G$ (see Figure C.2). Since the state $(G, \cdot)$ is a common ancestor of both complete states $(G, \theta)$ and $(G', \theta')$, we can write the subtrajectory balance conditions (C.4) as

$$F(G, \theta) P_B(G \mid \theta) P_F(G' \mid G) P_F(\theta' \mid G') = F(G', \theta') P_B(G' \mid \theta') P_B(G \mid G') P_F(\theta \mid G), \tag{C.5}$$

where we abuse the notation $P_B(G \mid \theta)$ again to denote $P_B((G, \cdot) \mid (G, \theta))$. In fact, since the complete state $(G, \theta) \in \mathcal{X}$ has only a single parent state $(G, \cdot)$, we necessarily have $P_B(G \mid \theta) = 1$ (and similarly for $(G', \theta')$). Furthermore, we can use the observation from (Deleu et al., 2022) to write the flow $F(G, \theta)$ of a complete state $(G, \theta)$ as a function of its reward

$$F(G, \theta) = \frac{R(G, \theta)}{P_F(s_f \mid (G, \theta))}. \tag{C.6}$$

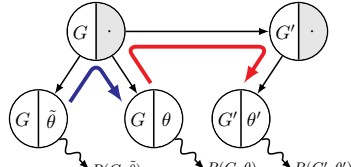

● Undirected paths of length 3
$$R(G', \theta')P_B(G \mid G')P_\phi(\theta \mid G) = R(G, \theta)P_\phi(G' \mid G)P_\phi(\theta' \mid G')$$

● Undirected paths of length 2 (differential form)
$$\nabla_\theta \log P_\phi(\theta \mid G) = \nabla_\theta \log R(G, \theta)$$

Figure C.2: Illustration of the undirected paths of length 3 (red) and of length 2 (blue) considered in this paper, and their corresponding subtrajectory balance conditions.

We can simplify (C.6) even further by observing that in the GFlowNet used here, $s_f$ is the *only* child of the complete state $(G, \theta) \in \mathcal{X}$. In other words, a complete state $(G, \theta)$ is not directly connected to any other $(G, \tilde{\theta})$; this is the (infinitely wide) tree structure rooted at $(G, \cdot)$ mentioned in Section 3.1. Since $s_f$ is the only child of $(G, \theta)$, we then necessarily have $P_F(s_f \mid (G, \theta)) = 1$, and therefore $F(G, \theta) = R(G, \theta)$. With these simplifications, (C.5) becomes

$$R(G, \theta)P_F(G' \mid G)P_F(\theta' \mid G') = R(G', \theta')P_B(G \mid G')P_F(\theta \mid G), \tag{C.7}$$

which is the subtrajectory balance condition in (6).

### C.3.2 Integrating undirected paths of length 2

Similar to Appendix C.3.1, we consider here an undirected of length 2 of the form $(G, \theta) \leftarrow (G, \cdot) \rightarrow (G, \tilde{\theta})$ (see Figure C.2). Since $(G, \cdot)$ is a common ancestor (a common parent in this case) of both complete states $(G, \theta)$ and $(G, \tilde{\theta})$, we can write the subtrajectory balance conditions (C.4) as

$$F(G, \theta)P_B(G \mid \theta)P_F(\tilde{\theta} \mid G) = F(G, \tilde{\theta})P_B(G \mid \tilde{\theta})P_F(\theta \mid G). \tag{C.8}$$

Using the same simplifications as in Appendix C.3.1 ($P_B(G \mid \theta) = P_B(G \mid \tilde{\theta}) = 1$), we get the following subtrajectory balance conditions for the undirected paths of length 2

$$R(G, \theta)P_F(\tilde{\theta} \mid G) = R(G, \tilde{\theta})P_F(\theta \mid G). \tag{C.9}$$

Note that these conditions are effectively redundant if the SubTB conditions over undirected paths of length 3 (6) are satisfied for all possible pairs of complete states $(G, \theta)$ and $(G', \theta')$. Indeed, if we write these conditions between $(G, \theta)$ and $(G', \theta')$ on the one hand, and between $(G, \tilde{\theta})$ and $(G', \theta')$ on the other hand (with a fixed $G'$ and $\theta'$)

$$R(G', \theta')P_B(G \mid G')P_F(\theta \mid G) = R(G, \theta)P_F(G' \mid G)P_F(\theta' \mid G) \tag{C.10}$$

$$R(G', \theta')P_B(G \mid G')P_F(\tilde{\theta} \mid G) = R(G, \tilde{\theta})P_F(G' \mid G)P_F(\theta' \mid G), \tag{C.11}$$

we get the same subtrajectory balance conditions over undirected paths of length 2 as in (C.9):

$$\frac{R(G, \theta)}{P_F(\theta \mid G)} = \frac{R(G', \theta')P_B(G \mid G')}{P_F(G' \mid G)P_F(\theta' \mid G')} = \frac{R(G, \tilde{\theta})}{P_F(\tilde{\theta} \mid G)}. \tag{C.12}$$

However, since the SubTB conditions (6) are only satisfied approximately in practice, it might be advantageous to also satisfy (C.9) in addition to those in (6). The equation above provides an alternative way to express (C.9). Indeed, (C.12) shows that the function

$$f_G(\theta) \triangleq \log R(G, \theta) - \log P_F(\theta \mid G) \tag{C.13}$$

is constant, albeit with a constant that depends on the graph $G$. Since this function is differentiable, this is equivalent to $\nabla_\theta f_G(\theta) = 0$, and therefore we get the differential form of the subtrajectory balance conditions in (7)

$$\nabla_\theta \log P_F(\theta \mid G) = \nabla_\theta \log R(G, \theta). \tag{C.14}$$

As we saw in Section 3.3, one way to enforce the SubTB conditions over undirected paths of length 3 is to create a learning objective that encourages these conditions to be satisfied, and optimizing it

using gradient methods. The learning objective has the form $\mathcal{L}(\phi) = \mathbb{E}_\pi[\tilde{\Delta}^2(\phi)]$, where $\tilde{\Delta}(\phi)$ is a non-linear residual term

$$\tilde{\Delta}(\phi) = \log \frac{R(G', \theta') P_B(G \mid G') P_\phi(\theta \mid G)}{R(G, \theta) P_\phi(G' \mid G) P_\phi(\theta' \mid G')}. \tag{C.15}$$

Suppose that the parameters $\phi$ of the GFlowNet are such that the subtrajectory balance conditions in (C.14) are satisfied for any $(G, \theta)$. Although this assumption is unlikely to be satisfied in practice, they will eventually be approximately satisfied over the course of optimization, given the discussion above about the relation between (C.9) and (C.7). Since $\theta$ and $\theta'$ depend on $\phi$ (via the reparametrization trick since they are sampled on-policy, see Section 5), taking the derivative of $\tilde{\Delta}^2(\phi)$, we get

$$\frac{d}{d\phi} \tilde{\Delta}^2(\phi) = \tilde{\Delta}(\phi) \cdot \frac{d}{d\phi} \big[ \log R(G', \theta') + \log P_\phi(\theta \mid G) \tag{C.16}$$
$$- \log R(G, \theta) - \log P_\phi(G' \mid G) - \log P_\phi(\theta' \mid G') \big].$$

Using the law of total derivatives, we have

$$\frac{d}{d\phi} \big[ \log P_\phi(\theta \mid G) - \log R(G, \theta) \big] = \underbrace{\left[ \frac{\partial}{\partial \theta} \log P_\phi(\theta \mid G) - \frac{\partial}{\partial \theta} \log R(G, \theta) \right]}_{= 0} \frac{d\theta}{d\phi} + \frac{\partial}{\partial \phi} \log P_\phi(\theta \mid G) \tag{C.17}$$

$$= \frac{\partial}{\partial \phi} \log P_\phi(\theta \mid G), \tag{C.18}$$

and similarly for the terms in $(G', \theta')$. The derivative of the objective then becomes

$$\frac{d}{d\phi} \tilde{\Delta}^2(\phi) = \tilde{\Delta}(\phi) \left[ \frac{\partial}{\partial \phi} \log P_\phi(\theta \mid G) - \frac{\partial}{\partial \phi} \log P_\phi(\theta' \mid G') - \frac{d}{d\phi} \log P_\phi(G' \mid G) \right]. \tag{C.19}$$

An alternative way to obtain the same derivative in (C.17) is to take $d\theta/d\phi = 0$ instead, meaning that we would not differentiate through $\theta$ (and $\theta'$). Using the stop-gradient operation $\perp$, this shows that the following objective

$$\mathcal{L}(\phi) \triangleq \mathbb{E}_\pi\big[\Delta(\phi)^2\big] = \mathbb{E}_\pi\left[ \left( \log \frac{R\big(G', \perp(\theta')\big) P_B(G \mid G') P_\phi\big(\perp(\theta) \mid G\big)}{R\big(G, \perp(\theta)\big) P_\phi(G' \mid G) P_\phi\big(\perp(\theta') \mid G'\big)} \right)^2 \right] \tag{C.20}$$

takes the same value and has the same gradient (C.19) as the objective in (C.15) when the subtrajectory balance conditions (in differential form) over undirected paths of length 2 are satisfied.

While optimizing (C.20) alone leads to eventually satisfying the subtrajectory balance conditions over undirected paths of length 2, it may be advantageous to explicitly encourage this behavior, especially in cases where $d$ is larger and/or for non-linear models. We can incorporate some penalty to the loss function, such as

$$\tilde{\mathcal{L}}(\phi) = \mathcal{L}(\phi) + \frac{\lambda}{2} \mathbb{E}_\pi\Big[ \big\| \nabla_\theta \log P_\phi(\theta \mid G) - \nabla_\theta \log R(G, \theta) \big\|^2 \tag{C.21}$$
$$+ \big\| \nabla_{\theta'} \log P_\phi(\theta' \mid G') - \nabla_{\theta'} \log R(G', \theta') \big\|^2 \Big]$$

### C.3.3 Marginal distribution over complete states

**Theorem 3.1.** *If the SubTB conditions in (6) are satisfied for all undirected paths of length 3 between any $(G, \theta)$ and $(G', \theta')$ of the form $(G, \theta) \leftarrow (G, \cdot) \rightarrow (G', \cdot) \rightarrow (G', \theta')$, then we have*

$$P_\phi^\top(G, \theta) \triangleq P_\phi(G \mid G_0) P_\phi(\theta \mid G) \propto R(G, \theta),$$

*where $P_\phi(G \mid G_0)$ is the marginal probability of reaching $G$ from the initial state $G_0$ with any (complete) trajectory $\tau = (G_0, G_1, \ldots, G_{T-1}, G)$:*

$$P_\phi(G \mid G_0) \triangleq \sum_{\tau: G_0 \rightsquigarrow G} \prod_{t=0}^{T-1} P_\phi(G_{t+1} \mid G_t),$$

*using the conventions $G_T = G$, and $P_\phi(G_0 \mid G_0) = 1$.*

*Proof.* We assume that the SubTB conditions are satisfied for all undirected paths of length 3 between any $(G, \theta)$ and $(G', \theta')$, that is

$$R(G', \theta')P_B(G \mid G')P_\phi(\theta \mid G) = R(G, \theta)P_\phi(G' \mid G)P_\phi(\theta' \mid G'). \tag{C.22}$$

Let $G \neq G_0$ be a fixed DAG different from the initial state, and $\theta$ a set of corresponding parameters. Let $\tau = (G_0, \ldots, G_{T-1}, G)$ be an arbitrary trajectory from $G_0$ to $G$, where we use the convention $G_T = G$. For any $t < T$, if $\theta_t$ is a fixed set of parameters associated with $G_t$, then the SubTB conditions above can written for every timestep as

$$R(G_{t+1}, \theta_{t+1})P_B(G_t \mid G_{t+1})P_\phi(\theta_t \mid G_t) = R(G_t, \theta_t)P_\phi(G_{t+1} \mid G_t)P_\phi(\theta_{t+1} \mid G_{t+1}), \tag{C.23}$$

again, using the convention $\theta_T = \theta$. Taking the product of the ratio between $P_\phi$ and $P_B$ over the trajectory $\tau$, we get

$$\prod_{t=0}^{T-1} \frac{P_\phi(G_{t+1} \mid G_t)}{P_B(G_t \mid G_{t+1})} = \prod_{t=0}^{T-1} \frac{P_\phi(\theta_t \mid G_t)R(G_{t+1}, \theta_{t+1})}{R(G_t, \theta_t)P_\phi(\theta_{t+1} \mid G_{t+1})} \tag{C.24}$$

$$= \frac{P_\phi(\theta_0 \mid G_0)R(G, \theta)}{R(G_0, \theta_0)P_\phi(\theta \mid G)} \tag{C.25}$$

Moreover, the backward transition probability $P_B$, defined only over the transitions of the GFlowNet, induces a distribution over the trajectories from $G_0$ to $G$ (Bengio et al., 2023), meaning that

$$\sum_{\tau: G_0 \rightsquigarrow G} \prod_{t=0}^{T-1} P_B(G_t \mid G_{t+1}) = 1. \tag{C.26}$$

Therefore, we have

$$P_\phi(G \mid G_0)P_\phi(\theta \mid G) = P_\phi(\theta \mid G) \left( \sum_{\tau: G_0 \rightsquigarrow G} \prod_{t=0}^{T-1} P_\phi(G_{t+1} \mid G_t) \right) \tag{C.27}$$

$$= \frac{P_\phi(\theta_0 \mid G_0)}{R(G_0, \theta_0)} R(G, \theta) \left( \sum_{\tau: G_0 \rightsquigarrow G} \prod_{t=0}^{T-1} P_B(G_t \mid G_{t+1}) \right) \tag{C.28}$$

$$= \frac{P_\phi(\theta_0 \mid G_0)}{R(G_0, \theta_0)} R(G, \theta) \tag{C.29}$$

We saw in Appendix C.3.2 that $P_\phi(\theta_0 \mid G_0)/R(G_0, \theta_0)$ is independent of the value of $\theta_0$ if the SubTB conditions are satisfied for all undirected paths of length 3 (see (C.12)). This concludes the proof: $P_\phi(G \mid G_0)P_\phi(\theta \mid G) \propto R(G, \theta)$. $\square$

### C.3.4 Mini-batch training

**Proposition 3.2.** *Suppose that $\mathcal{B}$ is a mini-batch of $M$ observations sampled uniformly at random from the dataset $\mathcal{D}$, and let $\widehat{\mathcal{L}}_\mathcal{B}(\phi)$ be the learning objective defined in Section 3.3, where the reward has been replaced by the estimate $\widehat{R}_\mathcal{B}(G, \theta)$ in (13). Then we have $\mathcal{L}(\phi) \leq \mathbb{E}_\mathcal{B}\big[\widehat{\mathcal{L}}_\mathcal{B}(\phi)\big]$.*

*Proof.* We will first show that $\log \widehat{R}_\mathcal{B}(G, \theta)$ defined in (13) is an unbiased estimate of the log-reward $\log R(G, \theta)$ under a uniform distribution of the mini-batches $\mathcal{B}$. We can observe that by conditional independence of the observations $\boldsymbol{x}^{(n)}$ given $G$ and $\theta$, we have

$$\log P(\mathcal{D} \mid \theta, G) = \sum_{n=1}^{N} \log P(\boldsymbol{x}^{(n)} \mid \theta, G) = N\mathbb{E}_{\boldsymbol{x}}\big[\log P(\boldsymbol{x} \mid \theta, G)\big], \tag{C.30}$$

where the expectation is taken wrt. the uniform distribution over the observations in $\mathcal{D}$. It is important to note that we can decompose the likelihood term as in (C.30) because the observations are mutually independent given $G$ *and* $\theta$; if we were only conditioning on $G$ (i.e., using the marginal likelihood,

as in (Deleu et al., 2022)), then those observations would not be conditionally independent in general. Similarly, we have

$$\mathbb{E}_{\mathcal{B}}\left[\sum_{\boldsymbol{x}^{(m)}\in\mathcal{B}}\log P(\boldsymbol{x}^{(m)}\mid\theta,G)\right] = M\mathbb{E}_{\boldsymbol{x}}\left[\log P(\boldsymbol{x}\mid\theta,G)\right]. \tag{C.31}$$

Therefore, it shows that the estimate of the log-reward is unbiased:

$$\mathbb{E}_{\mathcal{B}}\left[\log\widehat{R}_{\mathcal{B}}(G,\theta)\right] = \frac{N}{M}\mathbb{E}_{\mathcal{B}}\left[\sum_{\boldsymbol{x}^{(m)}\in\mathcal{B}}\log P(\boldsymbol{x}^{(m)}\mid\theta,G)\right] + \log P(\theta\mid G) + \log P(G) \tag{C.32}$$

$$= \log P(\mathcal{D}\mid\theta,G) + \log P(\theta\mid G) + \log P(G) = \log R(G,\theta). \tag{C.33}$$

Recall that the estimate of the loss is defined as $\widehat{\mathcal{L}}_{\mathcal{B}}(\phi) = \mathbb{E}_{\pi}\left[\widehat{\Delta}_{\mathcal{B}}^2(\phi)\right]$, where the residual is defined by

$$\widehat{\Delta}_{\mathcal{B}}(\phi) = \log\frac{\widehat{R}_{\mathcal{B}}\big(G,\perp(\theta)\big)P_B(G\mid G')P_\phi\big(\perp(\theta)\mid G\big)}{\widehat{R}_{\mathcal{B}}\big(G',\perp(\theta')\big)P_\phi(G'\mid G)P_\phi\big(\perp(\theta')\mid G'\big)}. \tag{C.34}$$

Taking the expectation of this estimated loss wrt. a random mini-batch $\mathcal{B}$, we get

$$\mathbb{E}_{\mathcal{B}}\left[\widehat{\mathcal{L}}_{\mathcal{B}}(\phi)\right] = \mathbb{E}_{\pi}\left[\mathbb{E}_{\mathcal{B}}\left[\widehat{\Delta}_{\mathcal{B}}^2(\phi)\right]\right] \tag{C.35}$$

$$\geq \mathbb{E}_{\pi}\left[\mathbb{E}_{\mathcal{B}}\left[\widehat{\Delta}_{\mathcal{B}}(\phi)\right]^2\right] \tag{C.36}$$

$$= \mathbb{E}_{\pi}\left[\Delta^2(\phi)\right] \tag{C.37}$$

$$= \mathcal{L}(\phi), \tag{C.38}$$

where we used the convexity of the square function and Jensen's inequality in (C.36), and the unbiasedness of $\log\widehat{R}_{\mathcal{B}}(G,\theta)$ (as well as $\log\widehat{R}_{\mathcal{B}}(G',\theta')$) in (C.37); recall that $\Delta(\phi)$ is given in (C.20) (see also (8)).  $\square$

**Proposition C.1.** *The mini-batch gradient estimator is unbiased, i.e., $\nabla_\phi\mathcal{L}(\phi) = \mathbb{E}_{\mathcal{B}}\left[\nabla_\phi\widehat{\mathcal{L}}_{\mathcal{B}}(\phi)\right]$. Therefore, the local and global minima of the expected mini-batch loss coincide with those of the full-batch loss.*

*Proof.* We now show that the gradient estimator is unbiased. We observe that

$$\nabla_\phi\Delta(\phi) = \nabla_\phi\widehat{\Delta}_{\mathcal{B}}(\phi) \tag{C.39}$$

since only the terms corresponding to the rewards differ between $\Delta(\phi)$ and $\widehat{\Delta}_{\mathcal{B}}(\phi)$, and they do not depend on $\phi$. Therefore,

$$\mathbb{E}_{\mathcal{B}}\left[\nabla_\phi\widehat{\mathcal{L}}_{\mathcal{B}}(\phi)\right] = \mathbb{E}_{\mathcal{B}}\left[\nabla_\phi[\widehat{\Delta}_{\mathcal{B}}(\phi)^2]\right]$$
$$= 2\cdot\mathbb{E}_{\mathcal{B}}\left[\widehat{\Delta}_{\mathcal{B}}(\phi)\nabla_\phi\widehat{\Delta}_{\mathcal{B}}(\phi)\right]$$
$$= 2\cdot\mathbb{E}_{\mathcal{B}}\left[\widehat{\Delta}_{\mathcal{B}}(\phi)\right]\nabla_\phi\Delta(\phi)$$
$$= 2\Delta(\phi)\nabla_\phi\Delta(\phi)$$
$$= \nabla_\phi\left[\Delta(\phi)^2\right]$$
$$= \nabla_\phi\mathcal{L}(\phi),$$

as desired. This implies the expected mini-batch loss and full-batch loss differ by a constant and have the same set of local and global minima. This constant happens to equal

$$\text{Var}_{\mathcal{B}}\left[\log\frac{\widehat{R}_{\mathcal{B}}(G,\theta)}{\widehat{R}_{\mathcal{B}}(G',\theta')}\right],$$

and showing the difference equals this constant yields an alternative proof.  $\square$

---

**Algorithm 1** JSP-GFN training procedure

---

1: Initialize the trajectory at $G_0$ the empty graph
2: **repeat**
3:     ▷ *Interactions with the environment*
4:     **for** $K$ steps **do**
5:        Sample whether the trajectory continues or not: $a \sim P_\phi(\text{stop} \mid G_t)$
6:        **if** $a$ is the "stop" action **then**
7:           Reset the trajectory: $G_{t+1} = G_0$ is the empty graph
8:        **else**
9:           Sample a new graph $G_{t+1} \sim P_\phi(G_{t+1} \mid G_t, \neg\text{stop})$
10:          Store the transition $G_t \rightarrow G_{t+1}$ in the replay buffer
11:
12:     ▷ *Update of the parameters $\phi$*
13:     Sample a mini-batch $\mathcal{B}$ of transitions $G \rightarrow G'$ from the replay buffer
14:     Sample $\theta \sim P_\phi(\theta \mid G, \text{stop})$ (and similarly for $\theta'$) for each transition in $\mathcal{B}$
15:     Evaluate the rewards $R(G, \theta)$ and $R(G', \theta')$                            ▷ *(5)*
16:     Evaluate the loss $\mathcal{L}(\phi)$ based on $\mathcal{B}$                                     ▷ *(8)*
17:     $\phi \leftarrow \phi - \alpha \nabla_\phi \mathcal{L}(\phi)$        ▷ *Update the parameters with stochastic gradient methods*
18: **until** convergence criterion

---

# D    Additional experiments & experimental details

In addition to Bayesian structure learning methods based on variational inference (Lorch et al., 2021; Cundy et al., 2021) or GFlowNets (Nishikawa-Toomey et al., 2023), we also consider 2 baseline methods based on MCMC, and 2 methods based on bootstrapping (Friedman et al., 1999), as introduced in (Lorch et al., 2021). Metropolis-Hastings MC³ (MH-MC³) samples both graphical structures and parameters jointly at each move, whereas Metropolis-within-Gibbs MC³ (Gibbs-MC³) alternates between updates of the structure, and updates of the parameters; note that MC³ here refers to the Structure MCMC algorithm (Madigan et al., 1995). In terms of bootstrapping methods, we consider a variant (called B-GES*) based on GES (Chickering, 2002), and another (called B-PC*) based on PC (Spirtes et al., 2000). However, since this would yield an approximation of the marginal posterior $P(G \mid \mathcal{D})$ only, the parameter sample $\theta$ corresponding to a DAG $G$ correspond to the parameter inferred by $P(\theta \mid G, \mathcal{D})$ (see Appendix D.5.1).

## D.1   Sampling distribution

In Section 3.3, we saw that the learning objective of JSP-GFN can be written as $\mathcal{L}(\phi) = \mathbb{E}_\pi[\Delta^2(\phi)]$, where $\pi$ is a sampling distribution over $(G, \theta)$ and $(G', \theta')$ with full support. We use a combination of on-policy ($\pi = P_\phi$) and off-policy ($\pi$ is different from $P_\phi$) in order to train the GFlowNet: taking inspiration from (Deleu et al., 2022), transitions $G \rightarrow G'$ are sampled off-policy from a replay buffer, whereas their corresponding parameters $\theta$ and $\theta'$ are sampled on-policy using our current $P_\phi(\theta \mid G)$. Therefore, a key difference with Deleu et al. (2022) is that the reward $R(G, \theta)$ in (5) is calculated "lazily" when the loss is evaluated (i.e., only once $\theta$ and $\theta'$ are known), as opposed to being computed during the interaction with the state space and stored in the replay buffer alongside the transitions. Algorithm 1 gives a description of the training algorithm of JSP-GFN.

## D.2   Joint posterior over small graphs

### D.2.1   Data generation & modeling

**Data generation.**    We follow the same data generation process as in (Deleu et al., 2022; Lorch et al., 2021). More precisely, we first sample a graph from an Erdös-Rényi model (Erdős and Rényi, 1960) over $d = 5$ nodes, with $d$ edges on average (a setting typically referred to as ER1). Once the structure of the graph $G^\star$ is known, we sample the parameters $\theta^\star$ of the linear Gaussian model randomly from a standard Normal distribution $\mathcal{N}(0, 1)$. The linear Gaussian model is defined as

$$X_i = \sum_{X_j \in \text{Pa}_G(X_i)} \theta^\star_{ij} X_j + \varepsilon_i, \tag{D.1}$$

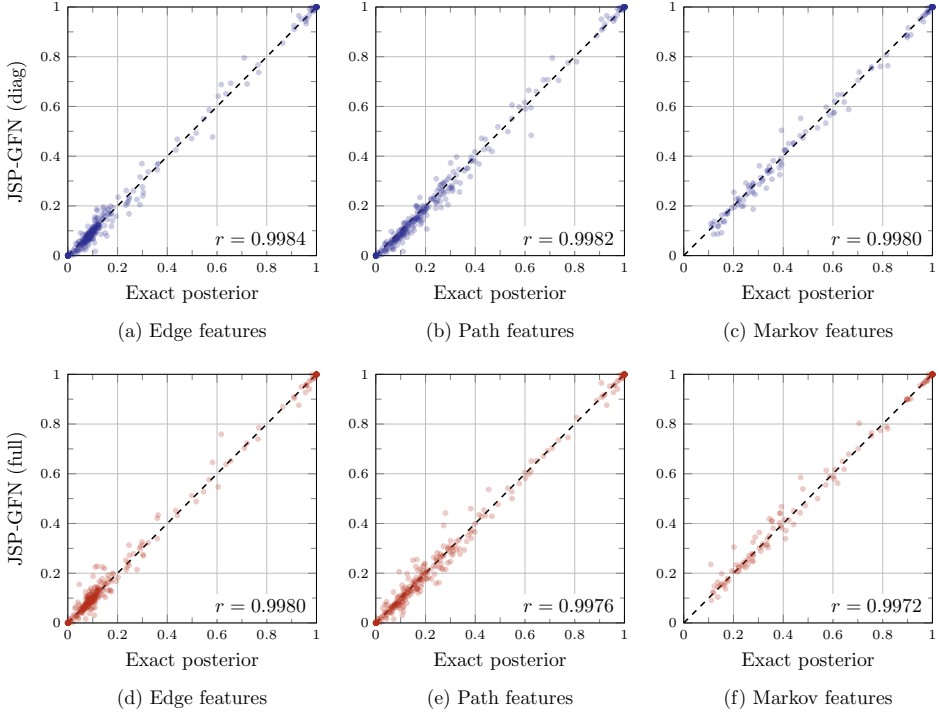

Figure D.1: Comparison of the marginals over graphs in terms of features computed with the exact posterior (x-axis) and the approximation given by JSP-GFN (y-axis). (a-c) Comparison with JSP-GFN (diag); (d-f) comparison with JSP-GFN (full). Each point corresponds to a pair of variables $(X_i, X_j)$ for each of the 20 datasets.

where $\theta_{ij}^{\star} \sim \mathcal{N}(0, 1)$, and $\varepsilon_i \sim \mathcal{N}(0, 0.01)$; this defines all the CPDs necessary for (1). Finally, we use ancestral sampling to generate $N = 100$ observations to create the dataset $\mathcal{D}$.

**Modeling.** We use a linear Gaussian Bayesian Network to model the data, where the CPD for the variable $X_i$ can be written as $P\big(X_i \mid \mathrm{Pa}_G(X_i); \theta_i\big) = \mathcal{N}(\mu_i, \sigma^2)$, where

$$\mu_i = \sum_{j=1}^{d} \mathbb{1}\big(X_j \in \mathrm{Pa}_G(X_i)\big)\theta_{ij}X_j, \tag{D.2}$$

and where $\sigma^2 = 0.01$ is a fixed variance across variables, matching the variance used for data generation. We place a unit Normal prior over the parameters $\theta_{ij}$ of the model: $P(\theta_{ij} \mid G) = \mathcal{N}(0, 1)$. This model differs from the widely used BGe score (Geiger and Heckerman, 1994) in that $\sigma^2$ is treated as a hyperparameter here, instead of a parameter of the model, and therefore the resulting log-reward is not *score equivalent* (i.e., placing the same reward for Markov equivalent DAGs; Koller and Friedman, 2009). We used a uniform prior over graphs. Under this model, the posterior distribution $P(\theta \mid G, \mathcal{D})$ can be computed analytically, and the proof is available in Appendix D.5.1.

### D.2.2 Comparing JSP-GFN with the exact posterior against features

In Section 5.1, we evaluated the accuracy of the posterior approximation returned by JSP-GFN by comparing them on the edge features, i.e., the marginal distribution of an edge $X_i \rightarrow X_j$ is present in the graph:

$$P(X_i \rightarrow X_j \mid \mathcal{D}) = \mathbb{E}_{P(G|\mathcal{D})}\big[\mathbb{1}(X_i \rightarrow X_j \in G)\big], \tag{D.3}$$

where the expectation in (D.3) is either over the true (marginal) posterior $P(G \mid \mathcal{D})$ (for the x-axis of Figure 2), or over the distribution $P_\phi^\top$ induced by the GFlowNet (discarding $\theta$, see Section 3.4, for the y-axis). Besides edge features, there exists other marginals of interest (Friedman and Koller, 2003; Deleu et al., 2022), such as the *path feature* and the *Markov feature*. The path feature corresponds to

Table D.1: Quantitative comparison between different Bayesian structure learning algorithms and the exact posterior on small graphs with $d = 5$ nodes. For each feature, the root meas-square error (RMSE) and Pearson's correlation coefficient between the features computed with the posterior approximation and the exact posterior are reported. Values are reported as the mean and $95\%$ confidence interval across 20 different datasets.

| | Edge features | | Path features | | Markov features | |
|---|---|---|---|---|---|---|
| | RMSE | Pearson's $r$ | RMSE | Pearson's $r$ | RMSE | Pearson's $r$ |
| MH-MC[3] | $0.357 \pm 0.022$ | $0.067 \pm 0.143$ | $0.368 \pm 0.027$ | $0.045 \pm 0.179$ | $0.341 \pm 0.017$ | $0.064 \pm 0.217$ |
| Gibbs-MC[3] | $0.357 \pm 0.022$ | $0.028 \pm 0.127$ | $0.367 \pm 0.026$ | $0.150 \pm 0.162$ | $0.341 \pm 0.018$ | $0.062 \pm 0.159$ |
| B-GES[*] | $0.263 \pm 0.070$ | $0.635 \pm 0.180$ | $0.302 \pm 0.080$ | $0.544 \pm 0.230$ | $0.129 \pm 0.022$ | $0.955 \pm 0.026$ |
| B-PC[*] | $0.305 \pm 0.057$ | $0.570 \pm 0.138$ | $0.349 \pm 0.058$ | $0.471 \pm 0.154$ | $0.354 \pm 0.072$ | $0.821 \pm 0.087$ |
| DiBS | $0.312 \pm 0.038$ | $0.737 \pm 0.071$ | $0.357 \pm 0.041$ | $0.710 \pm 0.079$ | $0.504 \pm 0.052$ | $0.643 \pm 0.093$ |
| BCD Nets | $0.215 \pm 0.055$ | $0.819 \pm 0.097$ | $0.266 \pm 0.057$ | $0.774 \pm 0.109$ | $0.327 \pm 0.040$ | $0.850 \pm 0.067$ |
| VBG | $0.237 \pm 0.037$ | $0.816 \pm 0.064$ | $0.284 \pm 0.027$ | $0.799 \pm 0.050$ | $0.434 \pm 0.058$ | $0.738 \pm 0.091$ |
| JSP-GFN (diag) | $\mathbf{0.018 \pm 0.005}$ | $\mathbf{0.998 \pm 0.001}$ | $\mathbf{0.022 \pm 0.005}$ | $\mathbf{0.998 \pm 0.001}$ | $\mathbf{0.019 \pm 0.006}$ | $\mathbf{0.999 \pm 0.001}$ |
| JSP-GFN (full) | $\mathbf{0.019 \pm 0.007}$ | $\mathbf{0.998 \pm 0.001}$ | $\mathbf{0.021 \pm 0.007}$ | $\mathbf{0.998 \pm 0.002}$ | $\mathbf{0.020 \pm 0.008}$ | $\mathbf{0.999 \pm 0.001}$ |

the marginal probability of a path $X_i \rightsquigarrow X_j$ being present in the graph, and the Markov feature is the marginal probability of a node $X_i$ being in the Markov blanket of $X_j$. In other words

$$P(X_i \rightsquigarrow X_j \mid \mathcal{D}) = \mathbb{E}_{P(G|\mathcal{D})}\big[\mathbb{1}(X_i \rightsquigarrow X_j \in G)\big] \tag{D.4}$$

$$P(X_i \in \text{MB}(X_j) \mid \mathcal{D}) = \mathbb{E}_{P(G|\mathcal{D})}\big[\mathbb{1}(X_i \in \text{MB}_G(X_j))\big], \tag{D.5}$$

where $\text{MB}_G(X_j)$ denotes the Markov blanket of $X_j$ in $G$. For the posterior approximation returned by JSP-GFN (and other methods), the expectations appearing in (D.4) & (D.5) are computed under the posterior approximation, and can be estimated using a Monte Carlo estimate over sample DAGs from the model.

In this experiment, we considered two variants of JSP-GFN. The first one, called *JSP-GFN (diag)*, where $P_\phi(\theta_i \mid G, \text{stop})$ in (12) is parametrized as a Normal distribution with a diagonal covariance matrix; this adds a modeling bias since here the exact posterior $P(\theta_i \mid G, \mathcal{D})$ is a Normal distribution with full covariance (see Appendix D.5.1). To control for this bias, the second model called *JSP-GFN (full)* assumes that (12) has a full covariance matrix, as in VBG (Nishikawa-Toomey et al., 2023).

In Figure D.1, we show a similar plot as in Figure 2 (a) for all these features, for both models JSP-GFN (diag) and JSP-GFN (full). We observe that for all features, the approximation of the posterior given by JSP-GFN is very accurate (as confirmed by the Pearson's correlation coefficients). Interestingly, the more expressive model JSP-GFN (full) seems to perform slightly worse than JSP-GFN (diag); this is also confirmed in part by the quantitative measures in Table D.1. This can be explained by the additional number of parameters of the neural network $\phi$ necessary to output the higher dimensional full covariance matrix (more precisely, a lower-triangular matrix corresponding to its Cholesky decomposition). In Table D.1, we show a quantitative comparison across the different Bayesian structure learning methods on the three features, in terms of RMSE and Pearson's correlation coefficient. Similar to Section 5.1, we observe that JSP-GFN provides a more accurate posterior approximation (at least in terms of its marginal over $G$) than other methods.

### D.2.3    Evaluation of the posterior approximations over parameters

In order to evaluate the quality of the posterior approximation over $\theta$, we measure how likely sample parameters from the approximation are under the exact posterior distribution $P(\theta \mid G, \mathcal{D})$. More precisely, we compute the cross-entropy between the posterior approximation $P_\phi(G, \theta)$ and the exact joint posterior $P(G, \theta \mid \mathcal{D})$: given a distribution $P_\phi(G, \theta)$ approximating the joint posterior $P(G, \theta \mid \mathcal{D})$, we estimate

$$-\mathbb{E}_{P_\phi(G,\theta)}\big[\log P(\theta \mid G, \mathcal{D})\big] \approx -\frac{1}{K}\sum_{k=1}^{K}\log P(\theta^{(k)} \mid G^{(k)}, \mathcal{D}), \tag{D.6}$$

where $\{(G^{(k)}, \theta^{(k)})\}_{k=1}^{K}$ are $K$ samples of the posterior approximation $P_\phi(G, \theta)$. We use this measure as it can be estimated from samples.

### D.3 Gaussian Bayesian Networks from simulated data

#### D.3.1 Data generation & modeling

**Data generation.** For the linear Gaussian experiment, the data generation process follows the process described for small graphs in Appendix D.2.1, except that we sample ground truth graphs $G^\star$ from an Erdös-Rényi model with $2d$ edges on average (a setting commonly referred to as ER2), for $d = 20$ variables.

For the non-linear Gaussian experiment, the data generation process is also similar, except that the CPDs are parametrized using a 2-layer MLP with 5 hidden units and a ReLU activation function (Lorch et al., 2021) with randomly generated weights. We also sample $N = 100$ observations to create the dataset $\mathcal{D}$.

**Modeling.** We use a Gaussian Bayesian Network to model the data, where the CPD for the variable $X_i$ can be written as $P(X_i \mid \mathrm{Pa}_G(X_i); \theta_i) = \mathcal{N}(\mu_i, \sigma^2)$, where

$$\mu_i = \mathrm{MLP}(\boldsymbol{M}_i \boldsymbol{X}; \theta_i), \tag{D.7}$$

where $\boldsymbol{M}_i = \mathrm{diag}\big(\mathbb{1}(X_1 \in \mathrm{Pa}_G(X_i)), \ldots, \mathbb{1}(X_d \in \mathrm{Pa}_G(X_i))\big)$ and $\boldsymbol{X} = (X_1, \ldots, X_d)$. The variance $\sigma^2 = 0.01$ is fixed across variables, and matches the variance used for data generation. Following Lorch et al. (2021) and matching the data generation process, we use a 2-layer MLP with 5 hidden units and a ReLU activation function, for a total of $|\theta| = 2,220$ parameters. The priors over parameters $P(\theta \mid G)$ and over graphs $P(G)$ follow the ones described in Appendix D.2.1.

#### D.3.2 Estimation of the log-terminating state probability

When the graphs are larger, it becomes impossible to compare the posterior approximation returned by JSP-GFN with the exact joint posterior $P(G, \theta \mid \mathcal{D})$ directly, since the latter becomes intractable (even with a linear Gaussian model) due to the super-exponential size of the sample space. Alternatively, since the terminating state probability $P_\phi^\top(G, \theta)$ of JSP-GFN should ideally be equal to the joint posterior (see Theorem 3.1), we have

$$\log P_\phi^\top(G, \theta) \approx \log P(G, \theta \mid \mathcal{D}) = \log R(G, \theta) - \log P(\mathcal{D}), \tag{D.8}$$

where $\log P(\mathcal{D})$ is a constant corresponding to the log-partition function. Therefore, we can compare the log-terminating state probability $\log P_\phi^\top(G, \theta)$ with the log-reward $\log R(G, \theta)$ (which we can compute analytically) for different samples $(G, \theta)$, and find a linear relation. This evaluation strategy was introduced in (Bengio et al., 2021).

However, recall from Theorem 3.1 that the terminating state probability is defined as

$$P_\phi^\top(G, \theta) = P_\phi(G \mid G_0) P_\phi(\theta \mid G) = P_\phi(\theta \mid G) \sum_{\tau: G_0 \rightsquigarrow G} \prod_{t=0}^{T-1} P_\phi(G_{t+1} \mid G_t), \tag{D.9}$$

where the summation is over all the possible trajectories $\tau = (G_0, G_1, \ldots, G_T)$ from $G_0$ to $G_T = G$. If $G$ is a DAG with $K$ edges, then there are $K!$ such trajectories (i.e., the $K$ edges could be added in any order), meaning that this sum is also intractable. We can leverage the fact that the backward transition probability $P_B(G_t \mid G_{t+1})$ induces a distribution over the trajectories $G_0 \rightsquigarrow G$ (Bengio et al., 2023) to write $P_\phi(G \mid G_0)$ as

$$P_\phi(G \mid G_0) = \sum_{\tau: G_0 \rightsquigarrow G} P_\phi(\tau) \tag{D.10}$$

$$= \sum_{\tau: G_0 \rightsquigarrow G} \frac{P_B(\tau)}{P_B(\tau)} P_\phi(\tau) \tag{D.11}$$

$$= K! \cdot \mathbb{E}_{\tau \sim P_B}\big[P_\phi(\tau)\big], \tag{D.12}$$

where $P_B(\tau) = 1/K!$, since the backward transition probability here is fixed to be uniform over parent states. This suggests a way to get an unbiased estimate of $P_\phi(G \mid G_0)$, hence of $P_\phi^\tau(G, \theta)$, based on Monte-Carlo estimation:

$$P_\phi(G \mid G_0) \approx \frac{K!}{M} \sum_{m=1}^{M} P_\phi(\tau^{(m)}), \tag{D.13}$$

where $\{\tau^{(m)}\}_{m=1}^M$ are trajectories from $G_0$ to $G$, sampled by removing one edge at time uniformly at random, starting at $G$ (i.e., following the backward transition probabilities $P_B$).

While (D.13) provides an unbiased estimate of $P_\phi(G \mid G_0)$, in practice the variance of this estimate will be large due to the combinatorially large space of trajectories, and therefore due to the wide range of values $P_\phi(\tau)$ may take. In order to reduce the variance, we can first identify some trajectories that would contribute the most to the sum in (D.10), and complement them with some randomly sampled trajectories as in (D.13). In other words, if we have access to a subset $\mathcal{T}_{\text{top}}$ of $B$ trajectories $\tau : G_0 \rightsquigarrow G$ that have a large $P_\phi(\tau)$, then

$$ P_\phi(G \mid G_0) = \sum_{\tau \in \mathcal{T}_{\text{top}}} P_\phi(\tau) + \sum_{\tau \notin \mathcal{T}_{\text{top}}} P_\phi(\tau) \approx \sum_{\tau \in \mathcal{T}_{\text{top}}} P_\phi(\tau) + \frac{K! - B}{M} \sum_{m=1}^M P_\phi(\tau^{(m)}), \quad \text{(D.14)} $$

where the sample trajectories $\{\tau^{(m)}\}_{m=1}^M$ can be obtained using rejection sampling, with a uniform proposal as above. The estimate in (D.14) is still unbiased, but with a lower variance. We can use beam-search to find the "top-scoring" trajectories in $\mathcal{T}_{\text{top}}$, with a beam-size $B$. More precisely, we need to run beam-search, starting at $G_0$, in such a way that the trajectories are guaranteed to end at $G$. We can achieve this by constraining the set of actions one can take at each step of expansion to move from a graph $G_t$ to $G_{t+1} = G_t \cup \{e\}$ (using this notation to denote that $G_{t+1}$ is the result of adding the edge $e$ to $G_t$), with the following score:

$$ \widetilde{P}_\phi(G_{t+1} \mid G_t) = \mathbb{1}(e \in G)P_\phi(G_{t+1} \mid G_t). \tag{D.15} $$

In other words, we only keep transitions corresponding to adding edges that are in $G$. Note that even though $\widetilde{P}_\phi$ is not a properly defined probability distribution (it does not sum to 1), we can still use this scoring function to run beam-search in order to find "top-scoring" trajectories.

### D.3.3 Additional comparisons with the ground-truth graphs

In Section 5.2, we compare JSP-GFN against other Bayesian structure learning in terms of their negative log-likelihood on held-out data. In addition to the negative log-likelihood though, there exists standard metrics in the structure learning literature that compare the posterior approximation with the ground truth graphs $G^\star$ used for data generation. For example, the *expected SHD*, that is estimated from graphs $\{G_1, \ldots, G_k\}$ sampled from the posterior approximation as

$$ \mathbb{E}-\text{SHD} \approx \frac{1}{n} \sum_{k=1}^n \text{SHD}(G_k, G^\star), \tag{D.16} $$

where $\text{SHD}(G, G^\star)$ counts the number of edges changes (adding, removing, reversing an edge) necessary to move from $G$ to $G^\star$. There is also the *area under the ROC curve* (AUROC) that compares the edge marginals estimated from the posterior approximation (i.e., the edge features, see Appendix D.2.2) and the target $G^\star$. We report these metrics in Figure D.2.

Although these metrics are used in the Bayesian structure learning literature, they also suffer from a number of drawbacks (Lorch et al., 2022). Namely, these metrics do not properly assess the quality of the posterior approximation (i.e., how close the approximation is the the true $P(G, \theta \mid \mathcal{D})$), but merely how close the sampled graphs are from $G^\star$. In general, and especially when the data is limited, the graphs sampled from the true posterior have no reason a priori to match exactly $G^\star$. Moreover, the expected SHD tends to favour overly sparse graphs on the one hand, or posterior approximations that collapse completely at $G^\star$ on the other hand, both situations indicating a poor approximation of the true $P(G, \theta \mid \mathcal{D})$.

### D.4 Learning biological structures from real data

#### D.4.1 Modeling

**Protein signaling networks from flow cytometry data.** Since the flow cytometry data has been discretized, we use a non-linear model with Categorical observations. The CPDs are parametrized using a 2-layer MLP with 16 hidden units and a ReLU activation function, i.e., $X_i \mid \text{Pa}_G(X_i) \sim \text{Categorical}(\pi_i)$, where

$$ \pi_i = \text{MLP}(\boldsymbol{M}_i \boldsymbol{X}; \theta_i), \tag{D.17} $$

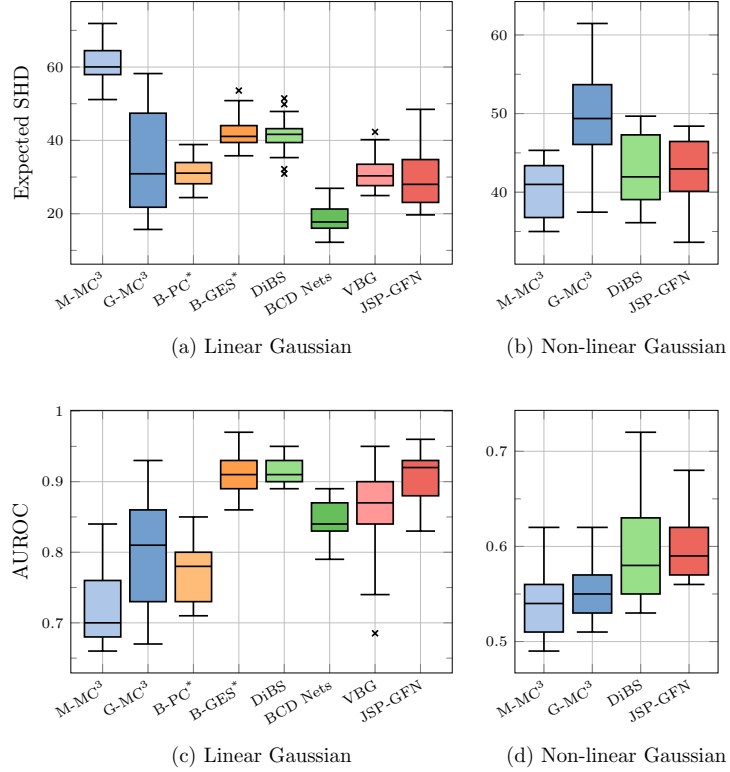

Figure D.2: (a-b) Comparison of JSP-GFN with other Bayesian structure learning methods in terms of the expected-SHD to the ground truth graphs $G^\star$ used for data generation. (c-d) Comparison in terms of Area Under the ROC curve (AUROC) to the ground truth graphs $G^\star$.

where $X$ encodes the discrete inputs as one-hot encoded values, and the MLP has a softmax activation function for the output layer. In total, the model has $|\theta| = 6,545$ parameters. The priors over parameters $P(\theta \mid G)$ and over graphs $P(G)$ follow the ones described in Appendix D.2.1.

**Gene regulatory networks from gene expression data.** Gene expression data is composed of either non-zero continuous data (when a gene is expressed) or (exactly) zero values (when the gene is inhibited). To capture this type of observations, we model CPDs of the Bayesian Network as zero-inflated Normal distributions:

$$P(X_i \mid \mathrm{Pa}_G(X_i); \theta_i) = \alpha_i \delta_0(X_i) + (1 - \alpha_i)\mathcal{N}(\mu_i, \sigma_i^2) \tag{D.18}$$

where $\mu_i$ is the result of a 2-layer MLP with 16 hidden units, as in (D.17). The parameters of the CPDs contain the parameters of the MLP, as well as the mixture parameter $\alpha_i$ and the variance of the observation noise $\sigma_i^2$, for a total of $|\theta| = 61,671$ parameters. The priors over parameters $P(\theta \mid G)$ and over graphs $P(G)$ follow the ones described in Appendix D.2.1.

### D.5 Proofs

#### D.5.1 Posterior of the linear Gaussian model

Recall that the CPD for the linear Gaussian model can be written as

$$P(X_i \mid \mathrm{Pa}_G(X_i); \theta_i) = \mathcal{N}(\mu_i, \sigma^2) \quad \text{where} \quad \mu_i = \sum_{j=1}^{d} \mathbb{1}(X_j \in \mathrm{Pa}_G(X_i))\theta_{ij}X_j, \tag{D.19}$$

and where $\sigma^2$ is a fixed hyperparameter. Moreover, we assume that the parameters have a unit Normal prior associated to them, meaning that

$$P(\theta_{ij} \mid G) = \begin{cases} \mathcal{N}(\mu_0, \sigma_0^2) & \text{if } X_j \to X_i \in G \\ \delta_0 & \text{otherwise,} \end{cases} \tag{D.20}$$

where $\mu_0 = 0$, $\sigma_0^2 = 1$, and $\delta_0$ is the Dirac measure at $0$, indicating that this parameter is always inactive.

We want to compute the posterior distribution $P(\theta_i \mid G, \mathcal{D})$; this is sufficient, since we know that the parameters of the different CPDs are mutually conditionally independent given $G$ and $\mathcal{D}$. Let $X \in \mathbb{R}^{N \times d}$ be the design matrix of the dataset $\mathcal{D}$ (i.e., the observations $\boldsymbol{x}^{(n)}$ concatenated row-wise), and by abuse of notation, we denote by $X_i$ the $i$th column of this design matrix. Let $D_i$ be a diagonal matrix, dependent on $G$, defined as

$$D_i = \mathrm{diag}\big(\mathbb{1}\big(X_1 \in \mathrm{Pa}_G(X_i)\big), \dots, \mathbb{1}\big(X_d \in \mathrm{Pa}_G(X_i)\big)\big). \tag{D.21}$$

We can rewrite the complete model above as

$$P(\theta_i \mid G) = \mathcal{N}(D_i\mu_0, \sigma_0^2 D_i) \tag{D.22}$$
$$P\big(X_i \mid \mathrm{Pa}_G(X_i); \theta_i\big) = \mathcal{N}(XD_i\theta_i, \sigma^2 I_N). \tag{D.23}$$

We abuse the notation above by treating a Dirac distribution at $0$ as the limiting case of a Normal distribution with variance $0$. Given this form, we can easily identify that the posterior over $\theta_i$ is a Normal distribution

$$P(\theta_i \mid G, \mathcal{D}) = \mathcal{N}(\bar{\mu}_i, \bar{\Sigma}_i) \qquad \text{where} \qquad \begin{aligned} \bar{\mu}_i &= \bar{\Sigma}_i \left[ \frac{1}{\sigma_0^2} D_i\mu_0 + \frac{1}{\sigma^2} D_i X^\top X_i \right] \\ \bar{\Sigma}_i^{-1} &= \frac{1}{\sigma_0^2} D_i^{-1} + \frac{1}{\sigma^2} D_i X^\top X D_i \end{aligned} \tag{D.24}$$

where we used the conventions $1/0 = \infty$ and $0 \times \infty = 0$. The masked entries of $D_i$ will correspond to zeroed-out entries in $\bar{\mu}_i$, and to rows and columns of $\bar{\Sigma}_i$ being equal to zero, effectively reducing the dimensionality of the distribution to the number of parents of $X_i$ in $G$ (e.g., this has an impact on the normalization constant of this distribution). In the limit case where $X_i$ has no parent in $G$, we recover $P(\theta_i \mid G, \mathcal{D}) = \delta_0$.

