$$\geq \mathbb{E}_{\pi}\big[\mathbb{E}_{\mathcal{B}}\big[\widehat{\Delta}_{\mathcal{B}}(\phi)\big]^2\big] \tag{D.8}$$

$$= \mathbb{E}_{\pi}\big[\Delta^2(\phi)\big] \tag{D.9}$$

$$= \mathcal{L}(\phi), \tag{D.10}$$

where we used the convexity of the square function and Jensen's inequality in (D.8), and the unbiasedness of $\log \widehat{R}_{\mathcal{B}}(G,\theta)$ (as well as $\log \widehat{R}_{\mathcal{B}}(G',\theta')$) in (D.9); recall that $\Delta(\phi)$ is given in (C.20) (see also (8)). □

Note that in the proof of Proposition D.1 above, we only used the convexity of the square function to conclude, but no other property of this function; in practice, we use the Huber loss instead of the square loss for stability, which is also a convex function. In the case of the square loss, we can get a stronger result in terms of unbiasedness of the gradient estimator.

**Proposition D.2.** *The mini-batch gradient estimator is unbiased, i.e., $\nabla_{\phi}\mathcal{L}(\phi) = \mathbb{E}_{\mathcal{B}}\big[\nabla_{\phi}\widehat{\mathcal{L}}_{\mathcal{B}}(\phi)\big]$. Therefore, the local and global minima of the expected mini-batch loss coincide with those of the full-batch loss.*

*Proof.* We now show that the gradient estimator is unbiased. We observe that

$$\nabla_{\phi}\Delta(\phi) = \nabla_{\phi}\widehat{\Delta}_{\mathcal{B}}(\phi) \tag{D.11}$$

since only the terms corresponding to the rewards differ between $\Delta(\phi)$ and $\widehat{\Delta}_{\mathcal{B}}(\phi)$, and they do not depend on $\phi$. Therefore,

$$\begin{aligned}
\mathbb{E}_{\mathcal{B}}\big[\nabla_{\phi}\widehat{\mathcal{L}}_{\mathcal{B}}(\phi)\big] &= \mathbb{E}_{\mathcal{B}}\big[\nabla_{\phi}[\widehat{\Delta}_{\mathcal{B}}(\phi)^2]\big] \\