# OpenReview forum: "Joint Bayesian Inference of Graphical Structure and Parameters with a Single Generative Flow Network"
_NeurIPS.cc/2023/Conference — NeurIPS 2023 poster_

### Official Review · Reviewer_9gh3 · 2023-06-21

**Soundness:** 3 good
**Presentation:** 3 good
**Contribution:** 3 good
**Rating:** 6
**Confidence:** 3

**Summary:**

In a recent line of research, Generative Flow Networks (GNFs) are used for structure learning by drawing DAGs (that represent BN structures) from an implicit DAG distribution that fits the data (Deleu et al 2022). The contribution of the present paper is extending this line of research by drawing joint DAG and the parameters (of the factors associated with the DAG's nodes).
To be more precise, firstly they construct a graph progressively, by adding new links between nodes (as in Deleu et al 2022 paper), and then the parameters are chosen to form a pair of (structure, parameter).

**Strengths:**

1. The problem that this paper addresses i.e. "structure learning" is an important task.
2. The paper is overall well written and the literature review is thorough.
3. The reported results are good.
4. This is an interesting paper but unfortunately, I believe its theoretical flaws are serious and have to be addressed before publishing the work.

**Weaknesses:**

1. The core theory behind this work, i.e. equation (6), seems fundamentally incorrect and has measure theoretic problems:
Detailed balance is sufficient for "fixed point condition" --which in this case means, the convergence of the probability of the terminal structure/parameters  (G, theta) to a posterior probability that is proportional to its reward R(G, theta)-- only if the dimensionality is fixed. When the sub-spaces have different dimensions, measure-theoretic problems rise due to mapping a single point (in a low-dimensional subspace) to infinite points (in a high-dimensional subspace).
The authors correctly mention (in line 242) that MCMC encounters trans-dimensional move problems.
However, this (as discussed in Green's Reversible Jump MCMC paper) is due to the detailed balance condition which is the foundation of the present work as well. As such, as far as I can say, the same issue should be addressed here too otherwise the correct convergence is not guaranteed.

2. The code is not provided. As such, the reported results are not reproducable.

**Questions:**

1. I would appreciate it if the authors could comment on the mentioned measure theoretic problem.

2. In lines 176-179, there is a confusing statement that I interpret as follows: "Although there is no guarantee that equation (6) guarantees the convergence but if we assume that (6) holds for all pairs (which I suppose, means if we assume that the dimensionality is fixed which is not (?)) then the convergence is guaranteed."
Could you please clarify these lines?

---

> ### Author Rebuttal · Authors · 2023-08-09
>
> We would like to thank the reviewer for their comments; we believe that checking the soundness of theoretical results when available is an integral part of the reviewing process, and we appreciate it that you took the time to do so. **However, we want to draw the attention of all reviewers and the Area Chairs to the fact that there is no theoretical flaw in our submission, including in the context raised by Reviewer 9gh3.**
>
> The complete measure-theoretical treatment of continuous GFlowNets can be found in (Lahlou et al., 2023), but we provide here some additional details in the specific context of this submission. For some DAG $G$, let $\Theta_{G}$ denote the measurable space of parameters $\theta$ associated with $G$ (with its Borel $\sigma$-algebra, which we omit here for clarity). **Note that the spaces $\Theta_{G}$ do not need to be the same for different graphs, and they may have different dimensionalities.** From this perspective, the reward $R(G, d\theta)$ and the forward transition probability $P_{\phi}(d\theta\mid G)$ are now **measures** over $\Theta_{G}$ (using infinitesimal notations).
>
> For any transition $G \rightarrow G'$ in the GFlowNet (i.e., $G'$ is the result of adding a single edge to $G$), Eq 6 of the submission represents **an equality between measures over the product space $\Theta_{G}\times \Theta_{G'}$**. Putting it differently, for any bounded measurable function $h: \Theta_{G}\times \Theta_{G'}\rightarrow \mathbb{R}$, the condition reads:
>
> $$
> \iint_{\Theta_{G}\times \Theta_{G'}}h(\theta, \theta')R(G', d\theta')P_{B}(G\mid G')P_{\phi}(d\theta\mid G) = \iint_{\Theta_{G}\times \Theta_{G'}}h(\theta, \theta')R(G, d\theta)P_{\phi}(G'\mid G)P_{\phi}(d\theta'\mid G')
> $$
>
> We would like to emphasize once again that this condition is valid even when $\Theta_{G}$ and $\Theta_{G'}$ do not have the same dimensionality, since this is an equality between measures over the product space (not either individual space $\Theta$). In (Lahlou et al., 2023), it was shown how similar conditions can be translated into constraints in terms of probability **densities**.
>
> One mild (implicit) assumption we make in this submission is that for any DAG $G$, there exists a (common base) measure $\mu_{G}$ over $\Theta_{G}$ such that $R(G, d\theta) \ll \mu_{G}(d\theta)$ and $P_{\phi}(d\theta\mid G) \ll \mu_{G}(d\theta)$ are both absolutely continuous wrt. $\mu_{G}$. This assumption is indeed mild because as mentioned in Section 3.4 of our submission, $P_{\phi}(d\theta\mid G)$ approximates the posterior distribution $P(d\theta\mid G, \mathcal{D})$ (up to a multiplicative constant), and $R(G, d\theta) = P(\mathcal{D}, G, d\theta)$. Using the notations $R(G, \theta)$ and $P_{\phi}(\theta\mid G)$ to denote their Radon-Nikodym derivatives wrt. $\mu_{G}$, Eq 6 corresponds to the equality between measures above, written as an equality in terms of their Radon-Nikodym derivatives:
>
> $$
> R(G', \theta')P_{B}(G\mid G')P_{\phi}(\theta\mid G) = R(G, \theta)P_{\phi}(G'\mid G)P_{\phi}(\theta'\mid G')
> $$
>
> All the results in the submission hold if we explicitly work with measures instead of their derivatives, regardless of the dimensionalities of $\Theta$. In particular Thm 3.1 of our submission (satisfying these conditions implies that the GFlowNet induces a distribution $\propto R$) still holds true. We provide a detailed proof from a measure theoretical perspective in a separate comment below for completeness.
>
> Unlike the detailed balance condition in MCMC, Eq 6 (which is *not* the detailed balance condition for Markov chains) does not define a fixed-point condition for an invariant distribution to match some target distribution (here, the joint posterior). The objective of the GFlowNet is to find a forward transition probability $P_{\phi}$ such that its marginal distribution over complete states (called the terminating state probability in line 186) matches the joint posterior, via satisfying Eq 6. $P_{\phi}$ itself is not the invariant distribution that would arise from a fixed-point condition like the detailed balance.
>
> The key difference with RJ-MCMC is that in the GFlowNet presented here, we never have to make trans-dimensional moves: once the first phase ends (Sec 3.1), we commit to a specific graph $G$, and therefore we can sample from a distribution over $\Theta_{G}$; then once this is done (end of the second phase), we can start over from the initial state $(G_{0}, \cdot)$. In other words, we never make any move from one space $\Theta_{G}$ (or more precisely $(\{G\}\times \Theta_{G})$) to another $\Theta_{G'}$ with possibly a different dimensionality, unlike in RJ-MCMC. That’s why there is no need for an invertible transformation across dimensions in our Eq 6.
>
> > *In lines 176-179, there is a confusing statement […] Could you please clarify these lines?*
> >
>
> We see where this confusion comes from, and we apologize for it. “The SubTB conditions” in line 176 refers to generic subtrajectory balance conditions, as described in App C.2, and not the ones specifically for our setting in Eq 6. What we meant was that satisfying the subtrajectory balance conditions of the form in Eq C.3 do not guarantee (in general) that the GFlowNet induces a distribution proportional to $R$. However, satisfying the SubTB conditions for the specific subtrajectories of the form $(G, \theta) \leftarrow (G, \cdot) \rightarrow (G', \cdot) \rightarrow (G', \theta')$ considered in Sec 3.2, corresponding to the conditions in Eq 6, do guarantee this result (Thm 3.1). We propose to replace lines 176-179 by:
>
> *Although there is no guarantee in general that satisfying **the SubTB conditions in (C.3) for arbitrary subtrajectories** would yield…*
>
> We hope that this clarified the misunderstanding about our theoretical results, and we strongly encourage the reviewer to update their score since this was the only major source of confusion.
>
> ---
>
> (Lahlou et al., 2023) Salem Lahlou, et al. A Theory of Continuous Generative Flow Networks. ICML 2023.

---

> > ### Author Response · Authors · 2023-08-14
> > **Proof of Theorem 3.1 from a measure theoretical perspective**
> >
> > *(In this comment, we give some details on how to prove Theorem 3.1 of our submission from a purely measure theoretical perspective, using the condition above)*
> >
> > The joint distribution over graphs & parameters is defined over the measurable union space $\mathcal{X} \triangleq \bigcup_{G\in\mathbf{G}}(\{G\}\times \Theta_{G})$, where $\mathbf{G}$ represents the (finite) set of all DAGs over $d$ nodes. We first can prove that if we have transitions $G \rightarrow G' \rightarrow G''$ in the GFlowNet (i.e., $G'$ is the result of adding one edge to $G$, and $G''$ is the result of adding one edge to $G'$), then for any bounded measurable function $h: \Theta_{G}\times \Theta_{G''}\rightarrow \mathbb{R}$, we have:
> >
> > $$
> > \iint_{\Theta_{G}\times \Theta_{G''}}h(\theta, \theta'')R(G'', d\theta'')P_{B}(G\mid G')P_{B}(G'\mid G'')P_{\phi}(d\theta\mid G) = \iint_{\Theta_{G}\times \Theta_{G''}}h(\theta, \theta'')R(G, d\theta)P_{\phi}(G''\mid G')P_{\phi}(G'\mid G)P_{\phi}(d\theta''\mid G'')
> > $$
> >
> > We can use the SubTB condition above, observing that $h(\theta, \theta'')$ is a bounded measurable function of $\Theta_{G'}\times \Theta_{G''}$ (constant on $\Theta_{G'}$ for a fixed $\theta$), and using Fubini-Tonelli’s theorem.
> >
> > $$\Bigg[\iint_{\Theta_{G}\times \Theta_{G''}} h(\theta, \theta'')R(G'', d\theta'')P_{B}(G\mid G')P_{B}(G'\mid G'')P_{\phi}(d\theta\mid G)\Bigg]\times\Bigg[\int_{\Theta_{G'}}P_{\phi}(d\theta'\mid G')\Bigg]$$
> > $$= \int_{\Theta_{G}}\Bigg[\iint_{\Theta_{G'}\times\Theta_{G''}}h(\theta, \theta'')R(G'', d\theta'')P_{B}(G'\mid G'')P_{\phi}(d\theta'\mid G')\Bigg]P_{B}(G\mid G')P_{\phi}(d\theta\mid G)$$
> > $$= \int_{\Theta_{G}}\Bigg[\iint_{\Theta_{G'}\times\Theta_{G''}}h(\theta, \theta'')R(G', d\theta')P_{\phi}(G''\mid G')P_{\phi}(d\theta''\mid G'')\Bigg]P_{B}(G\mid G')P_{\phi}(d\theta\mid G)$$
> > $$= \iint_{\Theta_{G}\times \Theta_{G'}}\Bigg[\int_{\Theta_{G''}}h(\theta, \theta'')P_{\phi}(G''\mid G')P_{\phi}(d\theta''\mid G'')\Bigg]R(G', d\theta')P_{B}(G\mid G')P_{\phi}(d\theta\mid G)$$
> >
> > The quantity inside the brackets is a bounded measurable function of $\Theta_{G}\times \Theta_{G'}$. We can therefore apply the SubTB condition above.
> >
> > $$= \iint_{\Theta_{G}\times\Theta_{G'}}\Bigg[\int_{\Theta_{G''}}h(\theta, \theta'')P_{\phi}(G''\mid G')P_{\phi}(d\theta''\mid G'')\Bigg]R(G, d\theta)P_{\phi}(G'\mid G)P_{\phi}(d\theta'\mid G')$$
> > $$= \Bigg[\iint_{\Theta_{G}\times\Theta_{G''}}h(\theta, \theta'')R(G, d\theta)P_{\phi}(G'\mid G)P_{\phi}(G''\mid G')P_{\phi}(d\theta''\mid G'')\Bigg]\times\Bigg[\int_{\Theta_{G'}}P_{\phi}(d\theta'\mid G')\Bigg]$$
> >
> > Which proves the equality above on the product space $\Theta_{G}\times\Theta_{G''}$. By induction, we can also prove that for a partial trajectory $G_{0} \rightarrow G_{1}\rightarrow \ldots \rightarrow G_{T}$ in the GFlowNet, we have for any bounded measurable function $h: \Theta_{G_{0}}\times \Theta_{G_{T}}\rightarrow \mathbb{R}$:
> >
> > $$
> > \iint_{\Theta_{G_{0}}\times\Theta_{G_{T}}}h(\theta_{0}, \theta_{T})R(G_{T}, d\theta_{T})\prod_{t=0}^{T-1}P_{B}(G_{t}\mid G_{t+1})P_{\phi}(d\theta_{0}\mid G_{0}) = \iint_{\Theta_{G_{0}}\times\Theta_{G_{T}}}h(\theta_{0}, \theta_{T})R(G_{0}, d\theta_{0})\prod_{t=0}^{T-1}P_{\phi}(G_{t+1}\mid G_{t})P_{\phi}(d\theta_{T}\mid G_{T})
> > $$
> >
> > Theorem 3.1 of our submission can be rewritten in measure theoretical terms as
> >
> > > If the SubTB conditions above are satisfied for all undirected paths of length 3 between
> > any $(G, \theta)$ and $G', \theta')$ of the form $(G, \theta) \leftarrow (G, \cdot) \rightarrow (G', \cdot) \rightarrow (G', \theta')$, then we have for any bounded measurable function $h: \Theta_{G}\rightarrow \mathbb{R}$
> > > $\displaystyle \int_{\Theta_{G}}h(\theta)P_{\phi}^{\top}(G, d\theta) \triangleq \int_{\Theta_{G}}h(\theta) \sum_{G_{0}\rightsquigarrow G}\prod_{t=0}^{T-1}P_{\phi}(G_{t+1}\mid G_{t})P_{\phi}(d\theta\mid G)\propto \int_{\Theta_{G}}h(\theta)R(G, d\theta)$
> >
> > And we can prove this using the lemma above. Since any bounded measurable function $h:\Theta_{G}\rightarrow \mathbb{R}$ is also a measurable function over $\Theta_{G_{0}}\times \Theta_{G}$ (it is constant wrt. $\theta_{0}$), we can directly apply the lemma above (using the notation $G = G_{T}$)
> >
> > $$\Bigg[\int_{\Theta_{G_{0}}}R(G_{0}, d\theta_{0})\Bigg]\times \Bigg[\int_{\Theta_{G}}h(\theta)P_{\phi}^{\top}(G,d\theta)\Bigg]$$
> > $$= \sum_{G_{0}\rightsquigarrow G}\iint_{\Theta_{G_{0}}\times\Theta_{G}}h(\theta)R(G_{0}, d\theta_{0})\prod_{t=0}^{T-1}P_{\phi}(G_{t+1}\mid G_{t})P_{\phi}(d\theta\mid G)$$
> > $$= \sum_{G_{0}\rightsquigarrow G}\iint_{\Theta_{G_{0}}\times\Theta_{G}}h(\theta)R(G, d\theta)\prod_{t=0}^{T-1}P_{B}(G_{t}\mid G_{t+1})P_{\phi}(d\theta_{0}\mid G_{0})$$
> > $$= \Bigg[\int_{\Theta_{G_{0}}}P_{\phi}(d\theta_{0}\mid G_{0})\Bigg]\times \Bigg[\int_{\Theta_{G}}h(\theta)R(G, d\theta)\sum_{G_{0}\rightsquigarrow G}\prod_{t=0}^{T-1}P_{B}(G_{t}\mid G_{t+1})\Bigg]$$
> > $$= \int_{\Theta_{G}}h(\theta)R(G, d\theta)$$

---

> > ### Comment · Area_Chair_xZwM · 2023-08-15
> > **Reviewer 9gh3: Reply to Potential Theoretical Flaw**
> >
> > Dear Reviewer 9gh3,
> >
> > The authors have replied to your main concern regarding a potential theoretical flaw (with measure-theoretic and trans-dimensional issues). Have their reply and their additional comments addressed this concern? Do these change your assessment of the paper?

---

### Official Review · Reviewer_jHMM · 2023-07-06

**Soundness:** 3 good
**Presentation:** 2 fair
**Contribution:** 3 good
**Rating:** 5
**Confidence:** 3

**Summary:**

The authors present a method for learning a posterior distribution over Bayesian networks using an extended version of GFlowNets that can jointly construct the underlying DAG and associated parameters in a two-stage process. They use an objective based on a generalization of detailed balance to train their flow network and provide a theoretical justification that optimizing this generalized objective leads to the desired distribution.
The method is evaluated on synthetic problems and gene regulatory networks, outperforming recent methods based on MCMC.


**Strengths:**

The problem of learning Bayesian networks is relevant to the community. The authors provide a deep theoretical framework for the presented problem and derive it thoroughly. The strength of the method is that it provides not just a point estimate, but a full posterior distribution over Bayesian networks and parameters in an amortized fashion.

**Weaknesses:**

- While the theoretical framework is explained in much detail, the actual implementation is quite opaque to me. The authors could spend more time in explaining the architecture and implementation details.
- “You are strongly encouraged to submit code that produces this result.” [NeurIPS guidelines].
- A simple educational example that is dissected in detail could help to better understand the framework.
- The experimental results are restricted to some performance metrics, but especially in the biological datasets, it would be interesting to dig deeper. Could JSP-GFN discover known gene regulatory networks? How do posterior predictives look compared to the real data? What role does interventional vs. observational data play?

- The experiments provide little intuition about the regime on which this method can be expected to work in terms of the number of training data, the number of nodes, and the number of parameters per node.




**Questions:**

- Fig. D.3c: is this the negative LL (lower=better) or just LL? Depending on the answer the interpretation of the results would change.
- What is the relationship between accuracy vs. number of training data?
- How does the method scale with the number of nodes? Can the authors provide some intuition about the size of Bayesian networks one might be able to learn with this method, also in relation to the number of training data?
- Why is the additional use of a graph neural network necessary compared to the architecture used in DAG-GFlowNet?
- The authors discuss how $P_{\phi}(\theta_i|G,stop)$ can be parameterized by a Normal/diagonal-Normal distribution, or a 2-layer MLP for each $\theta_i$. How do the authors ensure that correlations between $\theta$s of different $G_i$ can be learned if different MLPs are used for each $\theta_i$?


**Limitations:**

The authors only touch on some limitations and a more detailed discussion would strengthen the paper.

---

> ### Author Rebuttal · Authors · 2023-08-09
>
> We would like to thank the reviewer for their detailed comments. Due to limited space, we focus our responses to the Questions, and defer the responses to other points raised in the review in a separate comment below.
>
> > *Fig. D.3c: is this the negative LL (lower=better) or just LL? Depending on the answer the interpretation of the results would change.*
>
> This is indeed a typo, and we would like to thank you for pointing this out. The results reported in Figure D.3c are the log-likelihoods. We will update the values in the table to report the negative log-likelihoods for consistency with the rest of the paper.
>
> > *What is the relationship between accuracy vs. number of training data?*
>
> In terms of accuracy of the posterior approximation, we expect JSP-GFN (or any other Bayesian structure learning method) to provide a more accurate approximation of the posterior when the dataset is small. When the dataset is larger, there are some challenges that arise in JSP-GFN, which we detail below.
>
> > *How does the method scale with the number of nodes? Can the authors provide some intuition about the size of Bayesian networks one might be able to learn with this method, also in relation to the number of training data?*
>
> One limitation for scaling in terms of number of nodes is the size of the neural network that is used to parametrize the forward transition probability: larger graphs mean larger inputs and therefore it is more expensive from a computational perspective. That being said, using a graph network instead of a transformer as in DAG-GFlowNet offers an advantage here (see below).
>
> Another limitation when it comes to scaling, which is beyond the scope of this paper, is the problem of effective exploration when the state space becomes so large (recall that it is super exponential in $d$). For the experiments we ran in this paper (up to $d=61$ nodes), we found that a naive exploration strategy as described in the general comment (**Practical implementation**) was sufficient. However, we envision that this could become a bottleneck for much larger graphs, where the state space becomes too large. Since this is a challenge that arises from GFlowNets themselves, this is an issue that is better studied in isolation and not in the narrow context of Bayesian structure learning (e.g., inspiration can be taken from the RL literature).
>
> But from our experiments, we can confidently attest that JSP-GFN is an effective method to approximate the posterior distribution for up to $d=50$ nodes (which is typically the scale attained by prior works), and maybe pushing it to $d=100$ nodes.
>
> In terms of the size of the dataset, JSP-GFN shares some limitations that were discussed in DAG-GFlowNet: when the dataset becomes large, the posterior distribution becomes more concentrated around a few graphs (and parameters). Therefore, the neural network that parametrizes the forward transition probability must be able to handle large variations in outputs, which can be challenging. This comes from the fact that Equation 8 corresponds (roughly speaking) to a regression problem where the target is the delta score $\log R(G', \theta') - \log R(G, \theta)$: if the posterior is very “peaky”, then the delta-score may take values with large magnitude (positive or negative, it scales roughly with the size of $\mathcal{D}$). One standard practice in machine learning would be to normalize the target of the regression problem, but this is impossible here since this would change the nature of the distribution we are trying to approximate. One solution suggested by the authors of DAG-GFlowNet, which could be applied to JSP-GFN too, would be to take inspiration of simulated annealing and introduce a temperature parameter in the posterior distribution that will decrease over training.
>
> In practice, JSP-GFN can handle datasets up to a few thousands examples (in our experiments, we had up to $N=4200$ in the flow cytometry data experiment).
>
> > *Why is the additional use of a graph neural network necessary compared to the architecture used in DAG-GFlowNet?*
>
> The graph network is not necessary, and we could have used a similar transformer architecture as the authors of DAG-GFlowNet. Our choice was motivated by the fact that the graph network was a much simpler architecture, specifically created for graph inputs, without compromising in terms of performance. The transformer used in DAG-GFlowNet is bound to have a fixed input size of $d^2$, regardless of the number of edges in the input graph; in contrast, the input graphs in the graph networks will have at most $d(d-1)/2$ edges, and will often have much fewer edges in practice. This allows us to better scale in terms of the size of graphs, as opposed to using a transformer.
>
> > *The authors discuss how $P_{\phi}(\theta_{i} \mid G, stop)$ can be parameterized by a Normal/diagonal-Normal distribution, or a 2-layer MLP for each $\theta_{i}$. How do the authors ensure that correlations between $\theta$s of different $G_{i}$ can be learned if different MLPs are used for each $\theta_{i}$?*
>
> We (implicitly) assume in this paper that the priors over graphs $P(G)$ and over parameters $P(\theta\mid G)$ are modular, which is a standard assumption in the structure learning literature, meaning that
>
> $$P(G) = \prod_{i=1}^{d}P(G_{i}) \qquad \mathrm{and} \qquad P(\theta\mid G) = \prod_{i=1}^{d}P(\theta_{i}\mid G_{i})$$
>
> In practice, we used a uniform prior over graphs, and a unit Gaussian prior over parameters (e.g., lines 783-787). Under modularity assumption, it’s easy to see that the posterior over parameters also decomposes as $P(\theta\mid G, \mathcal{D}) = \prod_{i}P(\theta_{i}\mid G_{i}, \mathcal{D})$. This is in fact at the core of our proof in Appendix D.6.1. Since $P_{\phi}(\theta_{i}\mid G, stop)$ effectively approximates the posterior over parameters (as mentioned in lines 231-232), there is no need to model correlations between $\theta$s, since they do not exist in the true posterior.

---

> > ### Author Response · Authors · 2023-08-14
> > **Additional responses**
> >
> > *(This comment contains additional responses to the review, due to the limited space)*
> >
> > > *While the theoretical framework is explained in much detail, the actual implementation is quite opaque to me. The authors could spend more time in explaining the architecture and implementation details.*
> > >
> >
> > We invite you to read the general comment (**Practical implementation**). For the details of the neural network architecture, we could add some additional details in the appendix too, but we believe that they would not contribute to the overall description in the main text.
> >
> > > *A simple educational example that is dissected in detail could help to better understand the framework.*
> > >
> >
> > We were hoping that the illustration in Figure 1 (along with the caption, and the main text in Section 3.1) would be sufficient for understanding how sampling from the GFlowNet works; we plan to add pseudo-code to describe the training process of JSP-GFN (see also the general comment, **Practical implementation**).
> >
> > A more detailed example would require either (1) specifying the whole state space, which could be too cluttered, or (2) specify real values of the forward transition probability $P_{\phi}$ to illustrate the process, both of which would be challenging to illustrate easily. However we are open to any suggestion to make this part as clear as possible.
> >
> > > *“You are strongly encouraged to submit code that produces this result.” [NeurIPS guidelines].*
> > >
> >
> > We invite you to read the general comment (**Code**).
> >
> > > *The experimental results are restricted to some performance metrics, but especially in the biological datasets, it would be interesting to dig deeper. Could JSP-GFN discover known gene regulatory networks? How do posterior predictives look compared to the real data? What role does interventional vs. observational data play?*
> > >
> >
> > We completely agree with you, and an analysis of the networks found by JSP-GFN against known regulation pathways would be extremely interesting. We are using the negative log-likelihood on held-out interventions to follow the evaluation of (Sethuraman et al., 2023), which introduced the dataset we are using here (specifically, the subset of genes). We would be cautious with interpreting the gene regulatory networks found by our method since they assume that the graphs are acyclic, thus ignoring potential feedbacks existing in real biological systems; see also Appendix B.2 for a discussion of these limitations for biological data.
> >
> > > *The authors only touch on some limitations and a more detailed discussion would strengthen the paper.*
> > >
> >
> > We can suggest adding the limitations highlighted above in terms of challenges for scaling the size of the graphs and size of datasets, to complement the limitations already mentioned in Appendix B.2.
> >
> > ---
> >
> > (Sethuraman et al., 2023) Muralikrishnna G. Sethuraman, Romain Lopez, Rahul Mohan, Faramarz Fekri, Tommaso Biancalani, Jan-Christian Hütter. NODAGS-Flow: Nonlinear Cyclic Causal Structure Learning. AISTATS 2023.

---

> > > ### Comment · Reviewer_jHMM · 2023-08-15
> > >
> > > I thank the authors for their response and I appreciate their additional explanations (although ignoring the NeurIPS guidelines of 6000 characters several times).
> > >
> > > As pointed out by several reviewers, the current manuscript needs further clarifications not only in terms of the practical implementation but also on the overall comprehensibility.
> > >
> > > If the authors don’t dare to interpret the results of the experiments on the real data, I don’t see any practical use in showing that we learned some abstract object which is not (?) useful to interpret. As such a (meaningful) application to real-world data is missing.
> > >
> > > Further, I find the (implicit) assumptions of modularity and thus independence of the corresponding posteriors rather strong and a major weakness. The authors should discuss this more clearly in their work.
> > >
> > > Given these comments and unsatisfactory responses on scalability, I will decrease my score but increase my confidence as I invested a reasonable amount of time to read the related literature and I am quite familiar with other approaches in this domain.
> > >
> > > *PS: I did not take into account Reviewer 9gh3 ‘s comment, and assume that there are no major theoretical flaws.*

---

> > > > ### Author Response · Authors · 2023-08-15
> > > >
> > > > Thank you for your response. We are quite surprised to hear that you are decreasing your score given our rebuttal.
> > > >
> > > > > *As pointed out by several reviewers, the current manuscript needs further clarifications not only in terms of the practical implementation but also on the overall comprehensibility.*
> > > >
> > > > We provided further explanations of the practical implementation in the general comment (**Practical implementation**), and suggested to include this in the final version of the paper. We would like to emphasize once again that all the components are already described in the paper. In terms of comprehensibility, we have addressed the points raised by Reviewer MmTz, and we welcome more precise aspects that you think may be updated to improve comprehensibility.
> > > >
> > > > > *Further, I find the (implicit) assumptions of modularity and thus independence of the corresponding posteriors rather strong and a major weakness. The authors should discuss this more clearly in their work.*
> > > >
> > > > The assumption of modularity is a widely used assumption in structure learning. Most of the score-based methods for structure learning make this exact assumption, under the name of *score decomposability*. For example in (Koller & Friedman, 2009):
> > > >
> > > > > Score decomposability has important ramifications when we search for structures that maximize the scores. The high-level intuition is that if we have a decomposable score, then a local change in the structure (such as adding an edge) does not change the score of other parts of the structure that remained the same. As we will see, the search algorithms we consider can exploit decomposability to reduce dramatically the computational overhead of evaluating different structures during search. *(Section 18.3.6.2)*
> > > > >
> > > > > Let $G$ be a network structure, let $P(G)$ be a structure prior satisfying structure modularity, and let $P(\theta_{G} | G)$ be a parameter prior satisfying global parameter independence and parameter modularity. Then, the Bayesian score over network structures is decomposable. *(Proposition 18.3)*
> > > >
> > > > And this assumption is made in an overwhelming majority of the score-based methods for structure learning. For example in the classic GES algorithm (Chickering, 2002):
> > > >
> > > > > For any scoring criterion $S(G, D)$, we say that $S$ is decomposable if it can be written as a sum of measures, each of which is a function only of one node and its parents. [...] Most scoring criteria derived in the literature are decomposable. [...] [W]e show how
> > > > using a decomposable scoring criterion leads to an efficient implementation of the two-phase greedy search algorithm.
> > > >
> > > > As far as we know, there is no well-established algorithm using a scoring function that is not satisfying this assumption. However, since you are quite familiar with the other approaches in this domain, it would be very helpful if you could share any work *not* making this modularity assumption.
> > > >
> > > > > *Given these comments and unsatisfactory responses on scalability*
> > > >
> > > > Can you give some details about what in our response regarding scalability is unsatisfactory? We have been as precise as possible in our response, showing the range of application of our method (backed up by our experiments), and giving insights as to why scaling even further would be challenging (both in terms of size of graphs and size of datasets). As far as we know, JSP-GFN scales to the same sizes of graphs and datasets as any other existing Bayesian structure learning method.
> > > >
> > > > ---
> > > >
> > > > (Koller & Friedman, 2009) Daphne Koller and Nir Friedman. Probabilistic Graphical Models: Principles and Techniques.
> > > >
> > > > (Chickering, 2002) David Maxwell Chickering. Optimal Structure Identification With Greedy Search. JMLR.

---

> > > > > ### Comment · Reviewer_jHMM · 2023-08-15
> > > > >
> > > > > Thank you for your fast response.
> > > > >
> > > > > I agree with the authors that they gave some explanation on the scalability, however, I have not seen any experimental evidence on the comments raised in the initial review.
> > > > >
> > > > >
> > > > > If I understood it correctly, JSP-GFN “offers an accurate approximation of the joint posterior […] on both simulated and real data.”
> > > > > While the power of JSP-GFN needs to be shown, on real data, I suspect that for many real-world problems the marginal posterior $P(\theta|G, D)$ might have non-neglectable correlations.
> > > > >
> > > > > Learning this marginal posterior can be seen as a classical inverse problem for a data-generating process $G$, which might be composed of several components $G_i$. For such a problem, all kinds of methods exist to learn non-modular (correlated) posterior distributions, starting from standard MCMC approaches for models with known likelihood, over ABC (Sisson et al. 2018) or more recent SBI approaches (Cranmer et al. 2020) fueled by normalizing flows (Papamakarios et al. 2021).
> > > > >
> > > > > While the authors highlight the to learn the joint posterior, they focus on the perspective of structural network learning.
> > > > >
> > > > > ---
> > > > >
> > > > > Sisson et al. Handbook of approximate Bayesian computation. CRC Press. (2018)
> > > > >
> > > > > Cranmer et al. The frontier of simulation-based inference. PNAS. (2020)
> > > > >
> > > > > Papamakarios et al. Normalizing flows for probabilistic modeling and inference. The Journal of Machine Learning Research. (2021)

---

> > > > > > ### Author Response · Authors · 2023-08-16
> > > > > >
> > > > > > We are indeed focusing on structure learning of Bayesian networks, and accordingly we are making standard assumptions in this field. Our goal with JSP-GFN is not to approximate the joint posterior between any data generating process $G$ and its parameters, but to approximate the joint posterior between the structure of a Bayesian network and the parameters of its CPDs (lines 5-7 in our submission: *"we propose in this paper to approximate the joint posterior over not only the structure of a Bayesian Network, but also the parameters of its conditional probability distributions."*). We understand that the methods linked above may be used for approximating non-modular posteriors, but the problems considered are *not* structure learning of Bayesian network. Once again, if you know any existing work *specifically on structure learning of Bayesian networks* that is not making this modular assumption, we would be happy to hear more about it.
> > > > > >
> > > > > > However, we would like to highlight that our framework *could* model correlations between different $\theta_{i}$s: we can model $P(\theta\mid G, stop)$ using any distribution, including non-factorized ones. That being said, just from a practical point of view this is not desirable: in our experiment on gene expression data, we have a total of $|\theta| = 61, 671$ (line 909), and having a model approximating $P(\theta\mid G, \mathcal{D})$ with a full $|\theta| \times |\theta|$ covariance matrix (e.g., if we approximate the conditional posterior over the parameters using a Normal distribution) means that the neural network predicting this covariance matrix would have to return $\approx 10^{9}$ outputs, which would be unreasonable. That's why the literature on Bayesian Neural Networks has focused on fully factorized posterior approximations. This is true for *any model* approximating $P(\theta\mid G, \mathcal{D})$, not specifically JSP-GFN.

---

> > > > > > > ### Comment · Reviewer_jHMM · 2023-08-18
> > > > > > >
> > > > > > > I apologize for my harsh critique about the modularity. I don’t know any structure learning method for Bayesian networks not making this assumption. But as the authors highlight, their method is more than structure learning.
> > > > > > >
> > > > > > > All other comments still hold.

---

### Official Review · Reviewer_c3AM · 2023-07-19

**Soundness:** 3 good
**Presentation:** 3 good
**Contribution:** 3 good
**Rating:** 6
**Confidence:** 2

**Summary:**

The authors utilize a generative flow network to jointly model the structure of the DAQ and its parameters.

**Strengths:**

The authors utilize GFlowNets to jointly model the structure of the DAQ and its parameters. The experiments on simulated data and real-world data are promising.

**Weaknesses:**

I am not an expert on this area and I did not bid this paper. So I will listen to other reviewers' opinion on weakness.

**Questions:**

None

**Limitations:**

The authors did not discuss this which I think they should.

---

> ### Author Rebuttal · Authors · 2023-08-09
>
> We would like to thank the reviewer for taking the time to read our submission, despite it not being in their area of expertise.
>
> The limitations of our work, along with broader impacts, are discussed in Appendix B.

---

### Official Review · Reviewer_5gCu · 2023-07-26

**Soundness:** 3 good
**Presentation:** 3 good
**Contribution:** 2 fair
**Rating:** 5
**Confidence:** 4

**Summary:**

This paper presents a new method for Bayesian structure learning from observational data, based on the framework of GFlowNets.

GFlowNets have been used for Bayesian structure learning, but previous approaches view the distributions over the parameters and structures of a model modular and may learn them with separate methods. The paper introduces a joint method that learns them with a single GFlowNet. To do so, the paper extends the previous approach (Deleu et al., 2022) by introducing a new reward function, deriving the subtrajectory balance for the new reward function, and introducing a few adaptations to the parametrization of the forward transition probabilities of GFlowNet.

The paper conducts experiments on small synthetic graphs to evaluate how well the posterior of the model of the graph and parameters matches the ground-truth. Moreover, experiments with data generated by linear/non-linear Gaussian Bayesian networks are also conducted, with a focus on evaluating different models' performance of predicting held-out data (likelihood).

**Strengths:**

1. Structure learning is an important area in machine learning. GFlowNets is a new and emerging framework for structure learning. There are only a few studies in this direction, which adds to the novelty of the paper.

2. The paper studies the problem of how to jointly learn parameters and structures in Bayesian structure learning with a single GFlowNet. The proposed solution to the problem is conceptually intuitive and technically sound, including the introduction of subtrajectory balance conditions and the parametrization of the forward transition probabilities.

3. The experiments of the paper are quite comprehensive. Benchmark datasets and recent advances are included. Multiple metrics are compared.

4. The quality of writing of the paper is good. The detailed derivations and settings are given. The paper is easy to follow.

**Weaknesses:**

1. Jointly learn parameters and structures in Bayesian structure learning with a single GFlowNet seems to be an interesting idea. But the benefit of doing so is less convincing in the paper. We have not seen theoretical or analytical study on why jointly learning them with one GFlowNet is useful. Empirically from the experiments of the paper, it is a bit hard to see that the joint learning scheme can help the quality of both the parameters and the structure. Specifically, the proposed method shows advantages on small graphs in Section 5.1, while its performance gain over others in Section 5.2 is a bit marginal in terms of the likelihood of held-out data. More importantly, one of the main purpose of structure learning is to reveal the true graph of the data, where the proposed method has not shown advantage over VBG (the method learns graphs with GFlowNet and parameters with variational inference, respectively) in Figure D.2. Therefore, it is a bit unclear whether people should learn them jointly.

2. The experiments in Figure 3 are about predicting held-out data. In the linear Gaussian case, BCD Nets shows very competitive performance. Given that BCD Nets and VBG are linear models, one can still use them to model data generated from non-linear Gaussian models and predict the held-out data, although it might be hard to use them to compare SHD or AUROC. These two methods may serve as informative baselines for the non-linear case (Figure 3.b).

**Questions:**

1. Is it possible to add the results of BCD Nets and VBG in the non-linear case of Figure 3.b?

2. Is it possible to provide complexity analysis between the proposed method and VBG?

---

> ### Author Rebuttal · Authors · 2023-08-09
>
> We would like to thank the reviewer for their detailed review and comments on our submission.
>
> > *But the benefit of doing so is less convincing in the paper. We have not seen theoretical or analytical study on why jointly learning them with one GFlowNet is useful.*
>
> We invite you to read the general comment (**Advantage of joint posterior**).
>
> > *Empirically from the experiments of the paper, it is a bit hard to see that the joint learning scheme can help the quality of both the parameters and the structure.*
>
> Just to be clear, all methods considered in our paper (both baseline methods, and our own method JSP-GFN) are jointly learning the parameters and structure. Our objective is not to show that finding the joint posterior over structures and parameters $P(G, \theta\mid D)$ is advantageous compared to the marginal posterior $P(G\mid D)$. Our comparison with the exact posterior in Sec 5.1 on small graphs show a clear advantage of JSP-GFN over existing methods, and this is confirmed on the downstream performance on larger graphs in Sec 5.2; the difference is statistically significant against most methods on linear Gaussian models. On non-linear Gaussian, the downstream performance alone (Fig 3b) is not enough to conclude, although JSP-GFN seems to offer some benefit. That’s why we complemented this analysis with a comparison with the log-joint probability in Fig 3c, to show the quality of the approximation. Note that this analysis with correlation plots in Fig 3c, which is a reliable metric for Bayesian structure learning as it compares with the exact posterior, cannot be done on other methods since they only provide samples of the posterior approximation, and the log-probability $\log Q(G, \theta)$ cannot be evaluated.
>
> > *More importantly, one of the main purpose of structure learning is to reveal the true graph of the data, where the proposed method has not shown advantage over VBG […] in Figure D.2. Therefore, it is a bit unclear whether people should learn them jointly.*
>
> We agree that the objective of structure learning is to recover the true graph generating the data, under identifiability assumptions. However, we respectfully disagree that this is the objective of **Bayesian** structure learning: the objective here is to provide a faithful approximation of the posterior distribution over graphs, not to find a single graph that generated the data. We argue in our submission that metrics such as SHD and AUROC (which are standard metrics in the structure learning literature) are not representative of the quality of the posterior approximation in the main text (lines 321-322), and expand on it in App D.4.3 (lines 886-893).
>
> One argument we give in the paper is that a method that would always sample the true graph would fare very well under these two metrics (with an expected SHD of 0 and an AUROC of 1), even though this would most likely be a very poor approximation of the posterior distribution $P(G\mid D)$. This would be a very good structure learning method, just not a good Bayesian structure learning one. That’s why we focused our analysis on metrics that do not compare with the ground truth graph (Figs 3 & D.3). As mentioned in our paper (lines 322-324), we included the expected SHD and AUROC in the Appendix only for completeness, since prior works on Bayesian structure learning still use these metrics.
>
> For the comparison with VBG specifically, VBG performs significantly worse than JSP-GFN in terms of negative log-likelihood on held-out data in Fig 3a. But beyond performance, JSP-GFN offers three advantages over VBG (some of which are summarized in Table A.1):
> - In its original formulation, VBG can only be applied to linear-Gaussian models. The authors mentioned that it is possible to apply VBG to non-linear CPDs, but this has not been theoretically or empirically validated yet (we mention this in our comparison with other methods in App A lines 503-504). On the other hand, JSP-GFN can naturally handle non-linear CPDs.
> - JSP-GFN is conceptually simpler than VBG, where training simply consists in minimizing the objective in Eq 8 using stochastic gradient methods. On the other hand, training VBG consists in an inner-loop/outer-loop procedure, where the GFlowNet needs to the re-trained on a new reward function during the inner-loop, and the reward function is updated during the outer-loop. This introduces a number of hyperparameters, which might be hard to tune (e.g., number of training iterations for the GFlowNet during the inner-loop).
> - It is impossible to use mini-batch training of the GFlowNet in VBG, due to the same limitations DAG-GFlowNet suffers from. This can limit the scope of application of VBG to cases where the size of the dataset is small.
>
> > *Is it possible to add the results of BCD Nets and VBG in the non-linear case of Figure 3.b?*
>
> Thank you for the suggestion. We only used methods that could be applied to non-linear models, to match the data generation process, following (Lorch et al., 2021).
>
> However, we followed your suggestion and evaluated both BCD Nets and VBG (methods based on linear Gaussian models) on our non-linear datasets as well. The results in terms of negative log-likelihood on held-out data are as follows (we’re reporting median & 25th/75th percentile here):
>
> | Algorithm | NLL (held-out) |
> | --- | --- |
> | M-MC3 | 2911.99 [1920.47, 3000.35] |
> | G-MC3 | 4224.05 [2982.29, 5385.90] |
> | DiBS | 3181.80 [2376.39, 4018.54] |
> | BCD Nets* | 2714.62 [1832.27, 2998.93] |
> | VBG* | 3030.28 [2311.04, 4805.90] |
> | JSP-GFN (ours) | 3235.74 [2327.23, 3429.66] |
>
> The rows with * denote the new results, and all other rows correspond to the results in Fig 3b. It is interesting to see that methods with linear Gaussian models perform well even on these datasets generated with non-linear functions, although the advantage is not statistically significant.
>
> ---
>
> (Lorch et al., 2021) Lars Lorch, et al. DiBS: Differentiable Bayesian Structure Learning. NeurIPS 2021.

---

> > ### Author Response · Authors · 2023-08-14
> > **Additional response regarding complexity analysis**
> >
> > *(This comment contains an addendum to our rebuttal, due to the limited space)*
> >
> > > *Is it possible to provide complexity analysis between the proposed method and VBG?*
> >
> > Complexity analysis might be difficult to derive since both of these methods involve training neural networks. However as described above, JSP-GFN is conceptually simpler than VBG as it only involves training a neural network with stochastic gradient methods, as opposed to VBG which requires a more complex inner-loop/outer-loop procedure, each inner-loop itself consisting in training a neural network.

---

> > > ### Comment · Reviewer_5gCu · 2023-08-17
> > > **Thanks for the response and additional experiments.**
> > >
> > > I've read the authors rebuttal and other reviewers' reviews. I saw a reviewer questioned the core theory of the detailed balance of the paper. But I'm not an expert in the particular theory so I look forward to seeing more discussions between the reviewer and the authors. Apart from that, I think the paper studies GFlowNet for causal discovery, which is an emerging and less studied area, although the idea is slightly incremental and the performance gain is a bit marginal. So I will keep my score.

---

### Official Review · Reviewer_rtyV · 2023-07-26

**Soundness:** 4 excellent
**Presentation:** 4 excellent
**Contribution:** 3 good
**Rating:** 8
**Confidence:** 2

**Summary:**

The present paper extends upon prior literature in making use of GFlowNets for structure learning.
The extension is concerned with finding an adequate approach to learning a complete Bayesian network, that is, the graph's structure and its parameters. Theory to support the consistency of the representation and empirical results comparing to state of the art.

**Strengths:**

_Note to the authors: I apologize in advance for being unable to provide an extensive, constructive feedback for your manuscript due to capacity issues on my end. Therefore, the confidence is being kept low. Also, there is an OpenReview bug regarding "First Time Reviewer," which sates 'Yes' although it is 'No.'_

* GFlowNets are an interesting alternative approach to sampling for various tasks of interest
* Learning a complete Graphical Structure is the optimum for structure learning
* Wonderful presentation, both theory and visualizations/schematics
* Self-enclosed work showcasing most/all relevant aspects to answering this research question

**Weaknesses:**

* 'Just' learning the parameters brings a completion to the structure learning approach for GFlowNets but IMHO poses a more marginal/incrimental improvement

**Questions:**

* What are the key assumptions (or 'sacrifices') one has to take for the real world example?

**Limitations:**

---

---

> ### Author Rebuttal · Authors · 2023-08-09
>
> We would like to thank the reviewer for their review and their enthusiasm regarding our submission.
>
> > *'Just' learning the parameters brings a completion to the structure learning approach for GFlowNets but IMHO poses a more marginal/incrimental improvement*
> >
>
> We invite you to read the general comment (**Advantage of joint posterior**).
>
> > *What are the key assumptions (or 'sacrifices') one has to take for the real world example?*
> >
>
> One major consideration when working with real data is to make sure that our modeling assumptions (how we parametrize the CPDs) match the statistics of the data as closely as possible. In particular, gene expression data initially consists of counts that can take a wide range of values, and a standard preprocessing technique consists in transforming this data through a $\log(1+x)$ function to make it so that the data “looks Gaussian”.
>
> However, since the raw data corresponds to counts, and a majority of those entries are 0 due to a lack of detection of a particular gene, this is also reflected after the transformation through $\log (1+x)$. Therefore, the processed data is not Gaussian, but closer to a mixture of a Gaussian and a Dirac at 0 (for these undetected genes): this is well modeled by a zero-inflated Normal distribution, as described in Equation D.29. Modeling this with only a Gaussian CPD (as is standard in the structure learning literature) would create a mismatch with the nature of the data.
>
> But even disregarding the heavy bias towards the value 0, since counts are positives, the function $\log(1+x)$ transforms the non-zero counts into “Gaussian-like” data with a positive mean. However when working with linear-Gaussian CPDs, it is often standard to work with normalized data, with mean 0, to avoid having to introduce a bias term in the linear function. Here, we can’t easily standardize data due to the Dirac at 0. Therefore, it can be advantageous to consider a more expressive function for the CPD, such as a MLP, to even offer the option to capture complex biological mechanisms. JSP-GFN has the advantage to be able to work with such non-linear models (parametrized by a MLP) with non-standard distributions (zero-inflated Normal).

---

> > ### Comment · Reviewer_rtyV · 2023-08-10
> > **Reponse to Author Rebuttal**
> >
> > Thank you for the response. I've read "Advantage of joint posterior" and the argument convinces me more so than before of the benefits of having a joint posterior. Furthermore, thank you for giving your perspective on necessary assumptions on the statsitics of the data, when the data is real world collected. I've also read the argument by Reviewer 9gh3 on the "measure theoretic flaw" and I disagree, that is, I'm siding with your detailled argument that resolves the alleged problem.

---

> > > ### Author Response · Authors · 2023-08-14
> > >
> > > Thank you very much for your reply. We are glad to hear that our responses were helpful.

---

### Official Review · Reviewer_MmTz · 2023-07-27

**Soundness:** 3 good
**Presentation:** 2 fair
**Contribution:** 3 good
**Rating:** 5
**Confidence:** 2

**Summary:**

This paper uses GFlowNets to learn a generative model that samples from the posterior distribution of both the graphical structure and corresponding parameterisation of Bayesian networks given some observation. The paper demonstrates that sub-trajectory balance conditions suffice to ensure that the GFlowNet induce a distribution that is well proportional to the reward function associated with a final sample. Experimental results demonstrate that this approach is able to recover that follows closely the true posterior distribution. The method outperform alternative strategies for linear gaussian CPD and is competitive with alternative approach for non-linear Gaussian.

**Strengths:**

- The problem statement and extension of Gflownets proposed to solve it are very sound. Jointly learning discrete structures and continuous parameterisation has many promising applications in various domains
- Although merging continuous and discrete GFlownets does not seem very original in itself, it requires to adapt the balancing conditions appropriately and making it work is a strong results in itself.
- Experimental results are very promising.

**Weaknesses:**

Although I have found the idea presented in the paper of learning a generative model over the structure and continuous parameters very interesting, I also find the paper hard to read and understand technically for a reader non-familiar with GFlowNets. For instance, it is unclear exactly how the policy is parameterised and optimised. Although section 3.3 discusses the learning objective it does not provide a clear explanation on how exactly the training process happens, which makes it hard for me to argue for more than a weak accept. I consider important for a paper at a top venue to be clear by itself and to not require to check all supplementary materials and or related work to be understood by someone with the sufficient mathematical background. I would suggest authors to merge sections 2 and 3 in order to save space and give a more precise description of the parameterisation of the policy and training algorithms used. Moreover, I find strange that 5.3 is also so weakly developed in the main materials whereas it hints very important capacity of the proposed method, which might handle well interventional data.

**Questions:**

- Why did not you try more expressive parameterisation of the CPD, such as with normalizing flows?
- Why did you only consider N=100 in 5.2?
- If the edge feature is a probability why do you used MSE, rather than a better metric for comparing distributions such as the KL?

**Limitations:**

/

---

> ### Author Rebuttal · Authors · 2023-08-09
>
> We would like to thank the reviewer for their review and their suggestions to improve the presentation of our method.
>
> > *I also find the paper hard to read and understand technically for a reader non-familiar with GFlowNets.*
> >
>
> We are sorry to hear that the presentation was not sufficiently clear. Our objective was precisely to make the description of GFlowNets as self-contained as possible in the paper, in particular in Section 2. We introduced all necessary notions in Section 2.2 with a description of how objects are sampled from the GFlowNet, and described the prior work DAG-GFlowNet in Section 2.3 since our submission builds on top of it. We also included additional details about training objectives in Appendix C.1 & C.2.
>
> > *For instance, it is unclear exactly how the policy is parameterised and optimised. Although section 3.3 discusses the learning objective it does not provide a clear explanation on how exactly the training process happens, which makes it hard for me to argue for more than a weak accept.*
> >
>
> We invite you to read the general rebuttal comment (**Practical implementation**) for a detailed response. We want to again emphasize that all the components were described in details in the main text.
>
> > *I consider important for a paper at a top venue to be clear by itself and to not require to check all supplementary materials and or related work to be understood by someone with the sufficient mathematical background.*
> >
>
> We absolutely agree with the reviewer here, and that’s why we strived for making this submission as self-contained as possible. However with only 9 pages available, we made our best effort to provide a description of GFlowNets as extensive as possible in the main text, and we necessarily have to defer details to the supplementary material, to leave enough space for a complete description of our contributions (Section 3).
>
> > *I would suggest authors to merge sections 2 and 3 in order to save space and give a more precise description of the parameterisation of the policy and training algorithms used.*
> >
>
> Section 2 discusses the necessary background for our submission (not our contributions), whereas Section 3 corresponds to our contributions, and as such we find that it would be confusing to merge these two sections to save space. Moreover, as mentioned above, the parametrization of the forward transition probability is described in great details in Section 3.4 already. What additional precisions would the reviewer like to be included?
>
> > *Moreover, I find strange that 5.3 is also so weakly developed in the main materials whereas it hints very important capacity of the proposed method, which might handle well interventional data.*
> >
>
> Due to limited space, we had to sacrifice the analysis of our experiments on real biological data for legibility, and defer them to Appendix D.5.2 (referenced in the main text in line 350). We only provided a description of the experiments and their relevance, in particular for interventional data. Upon acceptance and with an additional page in the camera ready version, we propose to move the complete Appendix D.5.2 (including Figure D.3) back into the main text.
>
> > *Why did not you try more expressive parameterisation of the CPD, such as with normalizing flows?*
> >
>
> Individually, each CPD may be simple (Gaussian, although with non-linear functions), but as a whole the Bayesian network can model quite a complex joint distribution. Using normalizing flows at the level of individual CPDs would be interesting, but it is not clear if this would significantly increase the expressiveness of the joint distribution. However, our method is not limited to Gaussian distributions (e.g., our experiments in Sec 5.3 on real data), and it could be applied to CPDs parametrized by normalizing flows too. We leave this as future work.
>
> > Why did you only consider N=100 in 5.2?
> >
>
> We used $N=100$ in our experiments in Section 5.2 in order to match the experimental setups used in a vast majority of the literature on Bayesian structure learning (Cundy et al., 2021; Lorch et al., 2021; Deleu et al., 2022; Wang et al., 2022). The motivation for a small dataset size is to highlight the advantage of the Bayesian perspective to capture uncertainty about the graphical structures.
>
> > *If the edge feature is a probability why do you used MSE, rather than a better metric for comparing distributions such as the KL?*
> >
>
> The edge features correspond to the marginal probability of a specific edge being in the graph, and is defined precisely in Equation D.4. This is not a probability *distribution* over edges, and therefore we can’t compare distributions using the KL divergence.
>
> What we could do though is compare the (marginal) distribution over graphs returned by the GFlowNet ($P_{\phi}^{\top}(G)$, using the notation of Theorem 3.1, discarding $\theta$ as described in Appendix D.3.2 line 794) and the exact marginal posterior $P(G\mid \mathcal{D})$ using the KL divergence (or any other divergence, such as the Jensen-Shannon divergence for symmetry): $\mathrm{KL}(P_{\phi}^{\top}||P(\cdot\mid \mathcal{D}))$. However, we wouldn’t be able to compute this metric for most of the baseline methods, since they only provide samples of the posterior approximation, and not the full distribution (we could instead use the cross-entropy metric, as we do for the component in $\theta$).
>
> ---
>
> (Cundy et al., 2021) Chris Cundy, et al. BCD Nets: Scalable Variational Approaches for Bayesian Causal Discovery. NeurIPS 2021.
>
> (Lorch et al., 2021) Lars Lorch, et al. DiBS: Differentiable Bayesian Structure Learning. NeurIPS 2021.
>
> (Deleu et al., 2022) Tristan Deleu, et al. Bayesian Structure Learning with Generative Flow Networks. UAI 2022.
>
> (Wang et al., 2022) Benjie Wang, et al. Tractable Uncertainty for Structure Learning. ICML 2022.

---

### Author Rebuttal · Authors · 2023-08-09

We would like to thank all the reviewers for taking the time to review our submission and for their valuable comments. While we are addressing specific questions in individual rebuttal comments, we would like to add a few points here which will be of interest for all reviewers and Area Chairs.

If any comment in the reviews has not been addressed in our individual rebuttal responses, this is due to the limited space available. We will provide these responses as soon as the discussion period starts.

### **Theory**

We would like to draw the attention of all reviewers and the Area Chairs here that **there is no theoretical flaw in our submission, including in the context raised by Reviewer 9gh3.**

### **Advantage of joint posterior**

We mentioned in the introduction of our submission (lines 37-46) why it is advantageous to consider the joint posterior $P(G, \theta\mid \mathcal{D})$ as opposed to the marginal posterior $P(G\mid \mathcal{D})$. However, as pointed out by reviewers rtyV & 5gCu, this could have been better highlighted in the paper.

To paraphrase what we said in lines 37-46, the main advantage of looking at $P(G, \theta\mid \mathcal{D})$ is that this allows us to consider Bayesian networks with more complex CPDs (e.g., non-linear CPDs, parametrized by a neural network). This has a significant impact in practice, since marginal posteriors are generally limited to linear Gaussian models (some exceptions exist, as mentioned in the paper in lines 32-36), and therefore approximating the joint posterior significantly increases the range of applications of Bayesian structure learning methods.

For the specific advantages of JSP-GFN over existing methods, we provide a table of comparison in Table A.1, where we compared our method to other methods based on variational inference and GFlowNets, on 7 different points: (1) if the method was approximating the joint posterior $P(G, \theta\mid \mathcal{D})$, (2) if the method could be applied to Bayesian networks with non-linear CPDs, (3) if the support of the distribution was guaranteed to be limited to the space of DAGs, (4) if the method could be applied to discrete data, (5) if the maximum number of parents for each node in the graph can be explicitly specified (common assumption in non-Bayesian structure learning), (6) if the method provides a method to sample from $P(G, \theta\mid \mathcal{D})$, and (7) if the method could learn the variational approximation based on mini-batches of data.

To the best of our knowledge, JSP-GFN is the only variational method that can approximate the joint distribution $P(G, \theta\mid \mathcal{D})$ for Bayesian networks with non-linear CPDs (including discrete random variables), while guaranteeing that the support of the distribution is limited to DAGs, which is essential for Bayesian networks. As such, it constitutes a major development in Bayesian structure learning.

### **Practical implementation**

While we described extensively all the necessary components of our method JSP-GFN in our submission, we would like to thank reviewers MmTz & jHMM for pointing out that the description of the practical implementation could be improved.

The way the policy (more precisely, the forward transition probability $P_{\phi}$) is parametrized is precisely described in Section 3.4; only the exact architectures of the neural networks used are not detailed (number of layers, number of hidden units), but we believe that they do not contribute to the overall description. The learning objective is described in Section 3.3, and explicitly formulated in Equation 8. Moreover, the way the behavior policy $\pi$ (used in the objective in Equation 8) is again described in details in Appendix D.1 (referenced in line 193 in the main text).

Training the GFlowNet then simply consists of standard stochastic optimization:

1. Draw samples $(G, \theta, G', \theta') \sim \pi$ according to the behavior policy $\pi$ as described in Appendix D.1, to obtain a Monte Carlo estimate of Equation 8;
2. Compute the different forward transition probabilities $P_{\phi}(G'\mid G)$, $P_{\phi}(\theta\mid G)$ and $P_{\phi}(\theta'\mid G')$ with the parametrization detailed in Section 3.4;
3. Evaluate the objective in Equation 8 (Section 3.3), and optimize it using stochastic gradient methods.

We can add this as a pseudo-code in the Appendix of the paper (the space in the main text unfortunately does not allow us to put it there). For example in Appendix D, in a new section before Appendix D.3.

The last component that is not described in out paper is how the transitions $G\rightarrow G'$ are obtained before putting them in the replay buffer, as described in Appendix D.1 (for the policy $\pi$). This is using a simple exploration strategy using $\epsilon$-sampling: we transition in the GFlowNet following $P_{\phi}(G'\mid G)$ with probability $\epsilon$, and select a new edge to add (among the valid edges) uniformly at random with probability $1-\epsilon$. We will add this as part of the pseudo-code above.

### **Code release**

While the source-code was not included as part of the supplementary material, as mentioned by reviewers jHMM & 9gh3, we have since released the code publicly. A link to an anonymized version of the code is available [here](https://anonymous.4open.science/r/jsp-gfn-AD5D/), and a link to the repository will be added to the final version of the paper upon acceptance. Our code is largely based on the code released by (Deleu et al., 2022) available [here](https://github.com/tristandeleu/jax-dag-gflownet).

---

(Deleu et al., 2022) Tristan Deleu, António Góis, Chris Emezue, Mansi Rankawat, Simon Lacoste-Julien, Stefan Bauer, Yoshua Bengio. Bayesian Structure Learning with Generative Flow Networks. UAI 2022.

---

### Decision · Program_Chairs · 2023-09-21

**Decision:**

Accept (poster)

**Comment:**

This paper presents a method for Bayesian structure learning from observational data, based on the GFlowNets framework. It shows that sub-trajectory balance conditions suffice to ensure that the GFlowNets induce a distribution that is well proportional to the reward function associated with a final sample.

All the reviewers recognize the importance of the problem this paper is addressing and that the proposed approach is sound. Although the method may seem incremental, it is important to acknowledge that adjusting the theory in GFlowNets (re: balancing conditions) and the promising results are substantial contributions. Several issues were raised regarding (i)  clarity of the presentation and the background necessary on GFlowNets; (ii)  details of the implementation; (iii) reproducibility and code and (iv) a potential theoretical flaw (with measure-theoretic and trans-dimensional issues). All these have been resolved by the authors’ rebuttal.

Because of the low confidence in the original set of assigned reviewers at the initial stages of the reviewing process, I recruited more reviewers and we ended up with 6 reviews. Overall, I am confident this paper is worthy of presentation at NeurIPS.